# An essential cell-autonomous role for hepcidin in cardiac iron homeostasis

**Samira Lakhal-Littleton[1]\*, Magda Wolna[1], Yu Jin Chung[1], Helen C Christian[1], Lisa C Heather[1], Marcella Brescia[1], Vicky Ball[1], Rebeca Diaz[2], Ana Santos[2], Daniel Biggs[2], Kieran Clarke[1], Benjamin Davies[2], Peter A Robbins[1]**

[1]Department of Physiology, Anatomy and Genetics, University of Oxford, Oxford, United Kingdom; [2]Wellcome Trust Centre for Human Genetics, University of Oxford, Oxford, United Kingdom

**Abstract** Hepcidin is the master regulator of systemic iron homeostasis. Derived primarily from the liver, it inhibits the iron exporter ferroportin in the gut and spleen, the sites of iron absorption and recycling respectively. Recently, we demonstrated that ferroportin is also found in cardiomyocytes, and that its cardiac-specific deletion leads to fatal cardiac iron overload. Hepcidin is also expressed in cardiomyocytes, where its function remains unknown. To define the function of cardiomyocyte hepcidin, we generated mice with cardiomyocyte-specific deletion of hepcidin, or knock-in of hepcidin-resistant ferroportin. We find that while both models maintain normal systemic iron homeostasis, they nonetheless develop fatal contractile and metabolic dysfunction as a consequence of cardiomyocyte iron deficiency. These findings are the first demonstration of a cell-autonomous role for hepcidin in iron homeostasis. They raise the possibility that such function may also be important in other tissues that express both hepcidin and ferroportin, such as the kidney and the brain.

\*For correspondence: samira. lakhal-littleton@dpag.ox.ac.uk

**Competing interests:** The authors declare that no competing interests exist.

## Introduction

As a constituent of hemoproteins, iron-sulphur proteins and other functional groups, iron is essential for cellular functions. Conversely, excess iron participates in cytotoxic Fenton-type chemical reactions. Thus, both iron deficiency and iron overload are detrimental to the cell. Therefore, the healthy functioning of tissues requires tight control of intracellular iron levels. These in turn are dependent both on cellular homeostatic pathways controlling iron uptake, usage, and storage, and on systemic pathways controlling iron levels in the plasma. In mammals, cellular iron homeostasis is controlled by the Iron Regulatory Proteins IRPs. Intracellular iron levels control the degradation of IRP2 and the conformational switch that confers the RNA-binding function of IRP1. IRPs in turn control the levels of iron uptake proteins such as transferrin receptor 1 (TfR1) and divalent metal transporter (DMT1), and the iron storage protein ferritin (*Rouault and Klausner, 1997*; *Rouault, 2006*). Systemic iron homeostasis is controlled by the hepcidin/ferroportin axis at the sites of iron entry into the circulation. Ferroportin (FPN), which is encoded by the Solute Carrier Family 40 Member 1 (*Slc40a1*), is the only known mammalian iron export protein and mediates iron release into the circulation from duodenal enterocytes, splenic reticuloendothelial macrophages and hepatocytes, the sites of iron absorption, recycling and storage respectively (*Ganz, 2005*; *Donovan et al., 2005*). FPN-mediated iron release is antagonized by the hormone hepcidin, also known as hepcidin antimicrobial peptide (HAMP). Produced primarily in the liver, hepcidin binds to and induces internalization of FPN, thereby limiting iron release into the circulation and its availability to peripheral tissues (*Nemeth et al., 2004*; *Qiao et al., 2012*). The importance of the HAMP/FPN axis is illustrated by diseases of systemic iron overload such as hemochromatosis and β-thalassaemia, where hepcidin

**eLife digest** Many proteins inside cells require iron to work properly, and so this mineral is an essential part of the diets of most mammals. However, because too much iron in the body is also bad for health, mammals possess several proteins whose role is to maintain the balance of iron. Two proteins in particular, called hepcidin and ferroportin, are thought to be important in this process. Some ferroportin is found in the cells that line the gut (where iron is absorbed into the body) and is required to release this iron into the bloodstream. It is also found in the spleen, which is where iron is removed from old red blood cells so that it can be recycled. The liver produces hepcidin to control when ferroportin is active in the gut and spleen.

Both hepcidin and ferroportin are also found in heart cells. In 2015, a study reported that that heart ferroportin plays an important role in heart activity. However, it was not clear what role hepcidin plays in this organ.

Now, Lakhal-Littleton et al. – including many of the researchers from the previous work – have genetically engineered mice such that they specifically lacked heart hepcidin, or had a version of ferroportin in their heart that does not respond to hepcidin. The experiments show that these changes caused fatal heart failure in the mice because ferroportin releases iron from heart cells in an uncontrolled manner. Lakhal-Littleton et al. were able to prevent heart failure by injecting the animals with iron directly into the bloodstream. These findings show that hepcidin produced outside the liver has a role in controlling the levels of iron in the body's organs.

Other organs such as the brain, kidney and placenta all have their own forms of hepcidin and ferroportin; further work could investigate the roles of these proteins. Finally, another challenge for the future will be to test whether new drugs that are being developed to block or mimic hepcidin from the liver have the potential to treat heart conditions in humans.

production is impaired (*Camaschella, 2005*; *Musallam et al., 2012*), and in anaemia of chronic disease where hepcidin production is inappropriately elevated (*De Falco et al., 2013*; *Nemeth and Ganz, 2014*).

Other than the liver, hepcidin is also found in tissues with no recognized role in systemic iron homeostasis, including the heart (*Merle et al., 2007*), the brain (*McCarthy and Kosman, 2014*), the kidney (*Kulaksiz, 2005*) and the placenta (*Evans et al., 2011*). The function of this extra-hepatic hepcidin remains unknown, but one hypothesis is that it is involved in local iron control. Relevant to this hypothesis are our recent findings that FPN is also present in the heart, that it is essential for cardiomyocyte iron homeostasis and that its cardiomyocyte-specific deletion leads to fatal cardiac iron overload in mice (*Lakhal-Littleton et al., 2015*). Therefore, we hypothesised that cardiac HAMP regulates cardiac FPN, and that such regulation is important for local iron control and for cardiac function.

To test this hypothesis, we generated two novel mouse models; the first with a cardiomyocyte-specific deletion of the *Hamp* gene, and the second, with cardiomyocyte-specific knock-in of *Slc40a1 C326Y*, that encodes a FPN with intact iron export function but impaired HAMP binding (*Schimanski et al., 2005*; *Drakesmith et al., 2005*). We studied cardiac function and iron homeostasis longitudinally in both models and report that both develop fatal cardiac dysfunction and metabolic derangement as a consequence of cardiomyocyte iron deficiency. This occurs against a background of otherwise normal systemic iron homeostasis. Both cardiac dysfunction and metabolic derangement are prevented by intravenous iron supplementation.

Our findings demonstrate that, at least in the cardiomyocyte, endogenously-derived HAMP plays an essential role in cellular, rather than systemic iron homeostasis. It does this through the autocrine regulation of cardiomyocyte FPN. Disruption of this cardiac HAMP/FPN leads to fatal cardiac dysfunction.

Currently, there is considerable clinical interest in strategies that target the HAMP/FPN axis for the treatment of systemic iron overload and iron deficiency. Our findings suggest that these strategies may additionally alter cardiac iron homeostasis and function. Other than the heart, both HAMP and FPN are also found in the brain, kidney and placenta (*McCarthy and Kosman, 2014*;

*Kulaksiz, 2005*; *Evans et al., 2011*; *Rouault, 2013*; *Moos and Rosengren Nielsen, 2006*; *Bastin et al., 2006*; *Wolff et al., 2011*). A pertinent question is the extent to which our findings in the heart extend to those tissues.

## Results

### Hamp expression and regulation in the heart

Hamp mRNA levels were approximately 30 fold lower in the adult mouse heart than in the liver (*Figure 1A*). Next, we examined the regulation of cardiac Hamp mRNA and HAMP protein in response to dietary iron manipulation, having first established that this dietary manipulation altered cardiac and liver iron levels (*Figure 1—figure supplement 1*). In both tissues, Hamp mRNA levels were decreased by the iron-deficient diet (Fe 2–5 ppm) and increased by the iron-loaded diet (Fe 5000 ppm) (*Figure 1A*). At the protein level, while changes in hepatic HAMP protein mirrored changes in its mRNA levels, cardiac HAMP protein was increased by the iron-deficient diet and unaffected by the iron-loaded diet (*Figure 1B*).

To explore further the regulation of cardiac hepcidin by iron, we isolated primary adult cardiomyocytes from mice, then carried out a timecourse treatment with the iron chelator desferroxamine DFO, or with ferric citrate FAC. Under control conditions, Hamp mRNA was upregulated following addition of fresh cardiomyocyte culture medium (cardiomyocytes are cultured for 2 hr in serum-free medium prior to this). Relative to control cardiomyocytes at the respective timepoint, Hamp mRNA was increased by FAC from 4 hr of treatment, and decreased by DFO at 4, 8 and 16 hr of treatment (*Figure 1C*). HAMP protein, measured by ELISA in supernatants was also increased following addition of fresh medium. Relative to control cardiomyocytes at the respective timepoint, HAMP protein in supernatants was increased by DFO as early as 2 hr, but remained unchanged by FAC at all timepoints (*Figure 1D*). Thus, the direction of response of the Hamp mRNA and HAMP protein to iron levels in vitro mirrored the responses seen in vivo.

Next, we aimed to understand the mechanisms underlying increased HAMP secretion in DFO-treated cardiomyocytes. To this end, we investigated the role of the prohormone convertase Furin, which in hepatocytes, is required for cleavage of the propeptide and the release of the mature HAMP peptide (*Valore and Ganz, 2008*). Furin expression has been reported in the heart (*Beaubien et al., 1995*; *Ichiki et al., 2013*), and we also found that its expression was upregulated in the hearts of mice provided an iron-deficient diet and in cardiomyocytes treated with DFO (*Figure 1—figure supplement 2*). When we measured HAMP in supernatants of cardiomyocytes treated with the Furin inhibitor decanoyl-Arg-Val-Lys-Arg-chloromethylketone (CMK), we found no increase in HAMP release following DFO treatment (*Figure 1D*). We confirmed that Hamp mRNA levels in cardiomyocytes were not altered by CMK treatment (*Figure 1—figure supplement 3*). Together, these results indicate that Furin upregulation mediates increased HAMP secretion from iron-deficient cardiomyocytes.

Having established that hepcidin is found in cardiomyocytes, we then aimed to define its function. To this end, we generated cardiomyocyte-specific Hamp knockout mice $Hamp^{fl/fl};Myh6.Cre+$ by crossing in-house conditional Hamp floxed (fl) mice with mice transgenic for Cre recombinase under the control of cardiomyocyte-specific promoter Myosin Heavy Chain 6 (*Myh6.Cre+*). Hamp mRNA (*Figure 1E*) and HAMP protein (*Figure 1F*) in the hearts of $Hamp^{fl/fl};Myh6.Cre+$ mice were significantly reduced compared to the hearts of $Hamp^{fl/fl}$ controls. Furthermore, compared to $Hamp^{fl/fl}$ cardiomyocytes, levels of HAMP protein in the supernatants of cardiomyocytes from $Hamp^{fl/fl};Myh6.Cre+$ mice were either greatly reduced or undetectable, both under baseline conditions and following treatment with DFO or FAC (*Figure 1—figure supplement 4*). Near complete ablation of the cardiac Hamp mRNA and HAMP protein in $Hamp^{fl/fl};Myh6.Cre+$ mice confirmed that cardiomyocytes were the primary site of hepcidin expression in the heart. Liver Hamp mRNA (*Figure 1E*) and HAMP protein (*Figure 1F*) were not different between $Hamp^{fl/fl};Myh6.Cre+$ and $Hamp^{fl/fl}$ controls, consistent with the cardiac-specific nature of Hamp gene deletion.

Also consistent with this cardiac-specific deletion, $Hamp^{fl/fl};Myh6.Cre+$ mice had normal levels of liver iron stores and circulating markers of iron homeostasis when compared to $Hamp^{fl/fl}$ controls, demonstrating that loss of cardiac hepcidin did not affect systemic iron homeostasis. In addition,

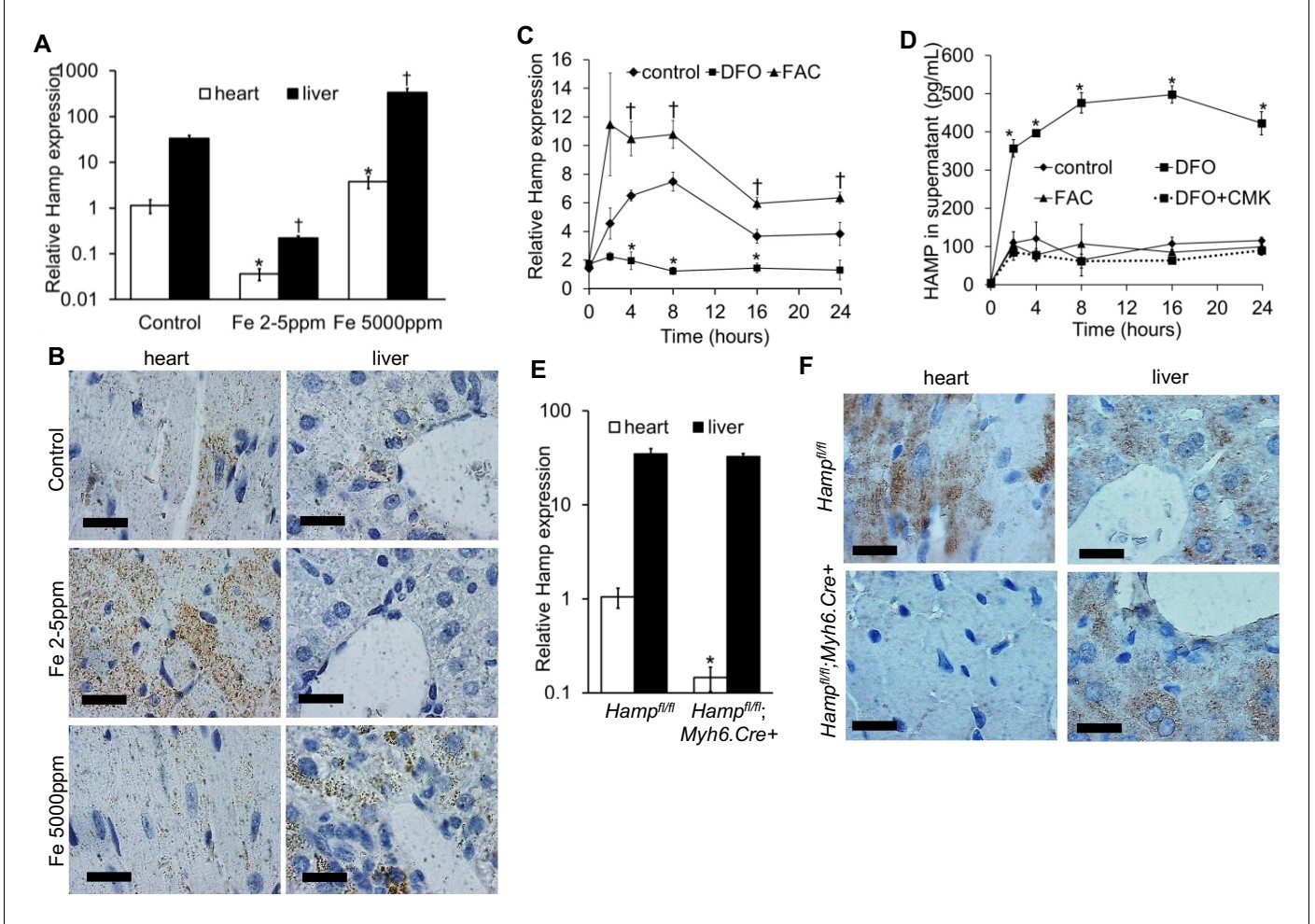

**Figure 1.** Hepcidin expression and regulation in the heart. (**A**) Relative *Hamp* mRNA expression in heart and liver of adult C57BL/6 mice, under control conditions and after provision of low or high iron diets. *p=0.047, 0.001 respectively relative to control hearts, †p = 0.006, 0.019 respectively relative to control livers. (**B**) Corresponding immunohistochemical staining for HAMP in heart and liver. (**C**) Relative *Hamp* mRNA expression in primary adult mouse cardiomyocytes cultured under control conditions or in presence of FAC or DFO. *p=0.023, 0.001 and 0.014 respectively relative to control. †p = 0.024, 0.037, 0.016 and 0.037 respectively relative to control at the same timepoint. (**D**) Corresponding HAMP protein levels in supernatants of primary cardiomyocytes. DFO treatment was carried alone (DFO) or presence of Furin inhibitor (DFO+CMK). *p=0.002, 0.020, 0.028, 0.014, 0.015 respectively relative to control at the same timepoint. (**E**) Relative *Hamp* expression in heart and liver of 3 month old *Hamp*^fl/fl and *Hamp*^fl/fl;*Myh6.Cre+* mice. *p=0.018 relative to cardiac *Hamp* in *Hamp*^fl/fl controls. (**F**) Corresponding immunohistochemical staining for HAMP in heart and liver. All values are plotted as mean ± SEM. Scale bar = 20 µm. n = 3 per group unless otherwise stated.

The following figure supplements are available for figure 1:

**Figure supplement 1.** Cardiac and liver iron following dietary iron manipulation.
**Figure supplement 2.** Furin regulation by iron.
**Figure supplement 3.** Relative *Hamp* mRNA expression in cardiomyocytes following treatment with Furin inhibitor CMK.
**Figure supplement 4.** HAMP in supernatants of primary cardiomyocytes.
**Figure supplement 5.** Confirmation of HAMP antibody specificity.
**Figure supplement 6.** HAMP detection by ELISA unaffected by FAC and DFO.

circulating HAMP levels were not reduced in the serum of *Hamp*[fl/fl];*Myh6.Cre+* mice, suggesting that cardiac hepcidin does not contribute significantly to circulating HAMP levels (**Table 1**).

## Fatal cardiac abnormalities in *Hamp*[fl/fl];*Myh6.Cre+* mice

To determine the effects of loss of cardiac hepcidin, we first assessed the cumulative survival of *Hamp*[fl/fl];*Myh6.Cre+* mice and *Hamp*[fl/fl] littermate controls over a period of 52 weeks. Significantly greater mortality was observed amongst *Hamp*[fl/fl];*Myh6.Cre+* mice, with only 29% of animals surviving to 52 weeks, compared with 90% of *Hamp*[fl/fl] controls. The median survival of *Hamp*[fl/fl];*Myh6.Cre+* mice was 28 weeks, whereas the majority of *Hamp*[fl/fl] controls were still alive at 52 weeks (**Figure 2A**).

Six-month old mice were sacrificed for assessment of cardiac morphology, which showed gross enlargement of the left ventricle (LV) in *Hamp*[fl/fl];*Myh6.Cre+* hearts compared to *Hamp*[fl/fl] controls (**Figure 2B**). Assessment of cardiomyocyte size by wheat germ agglutinin (WGA) staining confirmed that *Hamp*[fl/fl];*Myh6.Cre+* cardiomyocytes were hypertrophied (**Figure 2C–D**). This was accompanied by upregulation of expression of hypertrophic gene markers myosin heavy chain (*Myh7*) and natriuretic peptide precursor (*Nppb*) (**Figure 2E**). TUNEL staining for in-situ detection of cell death also showed significantly greater apoptosis in the hearts of *Hamp*[fl/fl];*Myh6.Cre+* mice than in *Hamp*[fl/fl] controls (**Figure 2F–G**).

To characterise further the phenotype caused by loss of cardiac hepcidin, we used cine MRI in anaesthetised *Hamp*[fl/fl];*Myh6.Cre+* mice and *Hamp*[fl/fl] littermate controls at 3, 6 and 9 months of age. Mid-ventricular cine MR images showed no differences between the two genotypes at 3 months of age. At 6 and 9 months of age, cine MR images showed marked enlargement of the LV in *Hamp*[fl/fl];*Myh6.Cre+* mice compared to *Hamp*[fl/fl] controls (**Figure 2H**). Formal quantitation of cardiac parameters by cine MRI confirmed enlargement of the LV lumen in *Hamp*[fl/fl];*Myh6.Cre+* mice, both at end-systole (LVES) (**Figure 2I**) and at end-diastole (LVED) (**Figure 2J**), accompanied by a decrease in LV ejection fraction (LVEF) from 62% to 42% (**Figure 2K**). Other parameters of cardiac performance were not significantly altered between mice from the two genotypes (**Table 2**). Taken together, histological examination of the hearts and cine MRI studies indicated that *Hamp*[fl/fl];*Myh6.Cre+* mice developed fatal LV dysfunction with reduced LVEF.

Such changes in cardiac performance could not be attributed to Cre recombinase toxicity in the heart as we have previously shown that *Myh6.Cre+* mice have normal cardiac function compared to wild type littermate controls (*Lakhal-Littleton et al., 2015*).

## The role of cardiomyocyte FPN in the phenotype of *Hamp*[fl/fl];*Myh6.Cre+* mice

We examined FPN protein in the hearts of *Hamp*[fl/fl];*Myh6.Cre+* mice, and found that FPN protein was markedly upregulated compared to *Hamp*[fl/fl] controls (**Figure 3A**), consistent with the idea that loss of cardiac HAMP was acting through upregulation of cardiomyocyte FPN. In order to test whether cardiac dysfunction arose from upregulation of cardiomyocyte FPN, we engineered mice where the *Slc40a1* gene harbours a conditional cardiac-specific C326Y point mutation, which confers HAMP-resistance while conserving the iron export function of FPN (*Schimanski et al., 2005*; *Drakesmith et al., 2005*). We confirmed that the *Slc40a1 C326Y* fl allele produced the C326Y

**Table 1.** Indices of systemic iron in six month old *Hamp*[fl/fl] and *Hamp*[fl/fl];*Myh6.Cre+* mice.
n = 6 per group. All values are shown as mean ± SEM.

| | *Hamp*[fl/fl] | *Hamp*[fl/fl];*Myh6.Cre+* |
|---|---|---|
| liver total elemental iron (ng/mg tissue) | 96.3 ± 12.2 | 88.7 ± 19.2 |
| liver ferritin (μg/mg total protein) | 0.65 ± 0.04 | 0.64 ± 0.05 |
| serum iron (μmol/L) | 28.60 ± 7.20 | 31.50 ± 8.40 |
| serum ferritin (mg/L) | 1.81 ± 0.04 | 1.88 ± 0.29 |
| hemoglobin (g/L) | 122.7 ± 11.5 | 116.0 ± 11.9 |
| serum hepcidin (μg/L) | 23.5 ± 7.6 | 23.9 ± 10.40 |

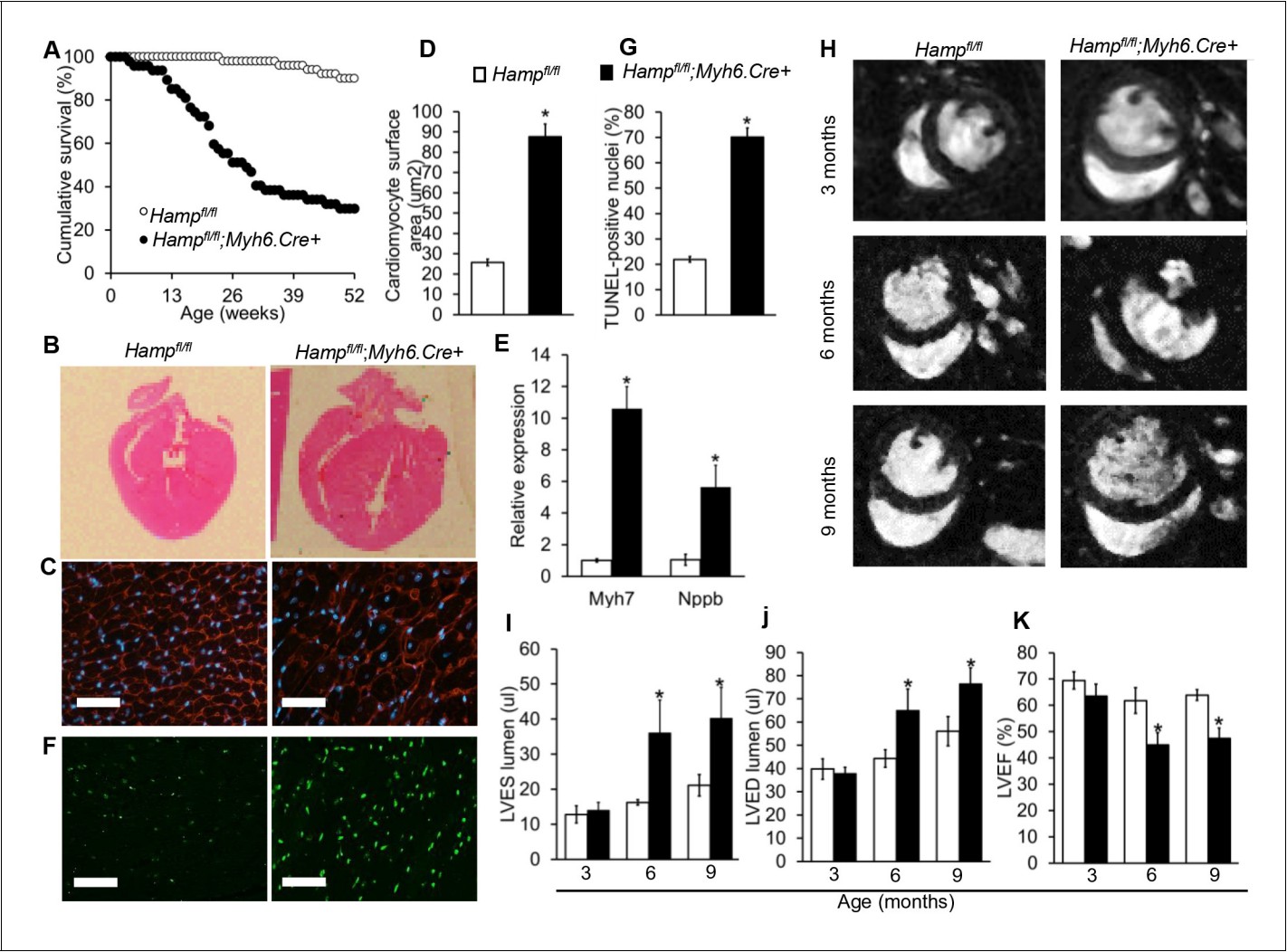

**Figure 2.** Fatal cardiac abnormalities in *Hamp*<sup>fl/fl</sup>;*Myh6.Cre+* mice. (**A**) Cumulative survival of *Hamp*<sup>fl/fl</sup>;*Myh6.Cre+* mice (n = 50) and *Hamp*<sup>fl/fl</sup> littermate controls (n = 47) over 52 weeks. (**B**) Representative H and E longitudinal heart sections from a six month old *Hamp*<sup>fl/fl</sup>;*Myh6.Cre+* mouse and *Hamp*<sup>fl/fl</sup> littermate control. (**C**) Representative WGA cardiac staining from a six month old *Hamp*<sup>fl/fl</sup>;*Myh6.Cre+* mouse and *Hamp*<sup>fl/fl</sup> littermate control. (**D**) Quantitation of cardiomyocyte size based on WGA staining in six month old *Hamp*<sup>fl/fl</sup>;*Myh6.Cre+* mice and *Hamp*<sup>fl/fl</sup> littermate controls. *p=0.001 relative to *Hamp*<sup>fl/fl</sup> littermate controls. (**E**) Relative expression of the hypertrophic gene markers *Myh7* and *Nppb* in hearts of 6 month old *Hamp*<sup>fl/fl</sup>;*Myh6.Cre+* mice and *Hamp*<sup>fl/fl</sup> littermate controls. *p=0.001, 0.047 for the respective gene relative to *Hamp*<sup>fl/fl</sup> littermate controls. (**F**) Representative cardiac in-situ TUNEL staining from a six month old *Hamp*<sup>fl/fl</sup>;*Myh6.Cre+* mouse and *Hamp*<sup>fl/fl</sup> littermate control. (**G**) Quantitation of percentage of apoptotic cardiomyocytes based on in-situ TUNEL staining in six month old *Hamp*<sup>fl/fl</sup>;*Myh6.Cre+* mice and *Hamp*<sup>fl/fl</sup> littermate controls *p=0.001 relative to *Hamp*<sup>fl/fl</sup> littermate controls. (**H**) Representative midventricular Cine MR images of hearts from *Hamp*<sup>fl/fl</sup>;*Myh6.Cre+* mice and *Hamp*<sup>fl/fl</sup> controls at 3, 6 and 9 months of age. (**I–K**) Cine MRI measurements of LV lumen, at end-systole (LVES), end-diastole (LVED), and of ejection fraction (LVEF) in *Hamp*<sup>fl/fl</sup>;*Myh6.Cre+* mice and *Hamp*<sup>fl/fl</sup> littermate controls at three months (n = 8 per group), six months (n = 11 per group, *p=0.043 for LVES, 0.047 for LVED and 0.020 for LVEF) and nine months (n = 5 per group, *p=0.044 for LVES, 0.042 for LVED and 0.034 for LVEF). p values are relative to *Hamp*<sup>fl/fl</sup> controls of the respective age. All Values are plotted as mean ± SEM. n = 3 per group unless otherwise stated. Scale bar = 50 µm.

The following source data is available for figure 2:

**Source data 1.** Source data file for *Figure 2I,J and K*.

transcript specifically in the heart (*Figure 3—figure supplement 1*) and that *Slc40a1 C326Y*<sup>fl/fl</sup>;*Myh6. Cre+* mice did not exhibit changes in systemic iron indices (*Table 3*). As seen in *Hamp*<sup>fl/fl</sup>;*Myh6.Cre+* mice, cardiomyocyte FPN was indeed upregulated in *Slc40a1 C326Y*<sup>fl/fl</sup>;*Myh6.Cre+* mice (*Figure 3B*).

**Table 2.** Non-LV parameters of cardiac function are not altered between *Hamp^fl/fl* and *Hamp^fl/fl;;Myh6.Cre+* mice.
Cine MRI measurements of cardiac function in *Hamp^fl/fl;Myh6.Cre+* mice and *Hamp^fl/fl* controls at three months (n = 8 per group), six months (n = 11 per group) and nine months (n = 5 per group) of age. Values are shown as mean ± SEM.

| | 3 months | | 6 months | | 9 months | |
|---|---|---|---|---|---|---|
| | *Hamp^fl/fl* | *Hamp^fl/fl; Myh6.Cre+* | *Hamp^fl/fl* | *Hamp^fl/fl; Myh6.Cre+* | *Hamp^fl/fl* | *Hamp^fl/fl; Myh6.Cre+* |
| Average mass (mg) | 70.51 ± 7.47 | 70.38 ± 5.33 | 72.59 ± 5.38 | 82.54 ± 11.32 | 78.30 ± 4.92 | 83.07 ± 5.44 |
| RVED lumen (µl) | 31.37 ± 2.55 | 26.81 ± 2.30 | 33.12 ± 3.19 | 30.63 ± 2.35 | 38.39 ± 3.88 | 39.95 ± 3.00 |
| RVES lumen (µl) | 7.43 ± 0.79 | 5.23 ± 0.67 | 8.70 ± 1.22 | 8.34 ± 1.06 | 13.36 ± 2.21 | 15.51 ± 2.47 |
| RVEF (%) | 76.42 ± 1.32 | 79.88 ± 2.80 | 73.66 ± 2.50 | 73.42 ± 2.22 | 65.20 ± 4.36 | 61.95 ± 3.53 |
| Stroke volume (µl) | 25.47 ± 1.99 | 22.73 ± 2.11 | 25.70 ± 2.70 | 23.76 ± 1.85 | 29.79 ± 2.30 | 29.84 ± 1.59 |
| Cardiac output (ml/min) | 10.34 ± 1.06 | 9.62 ± 0.90 | 10.72 ± 1.07 | 10.53 ± 1.05 | 11.46 ± 1.15 | 11.99 ± 1.38 |
| Heart Rate (bpm) | 404.07 ± 18.69 | 426.90 ± 17.94 | 419.64 ± 19.05 | 436.14 ± 14.30 | 384.84 ± 28.55 | 400.97 ± 41.20 |
| Heart/body weight ratio x1000 | 2.80 ± 0.18 | 3.09 ± 0.26 | 2.51 ± 0.23 | 2.98 ± 0.37 | 2.36 ± 0.28 | 2.88 ± 0.27 |

Like *Hamp^fl/fl;Myh6.Cre+* mice, *Slc40a1 C326Y^fl/fl;Myh6.Cre+* mice also had increased mortality relative to their littermate controls (**Figure 3C**). We then determined whether *Slc40a1 C326Y^fl/fl; Myh6.Cre+* mice developed the same phenotype of cardiac dysfunction as *Hamp^fl/fl;Myh6.Cre+* mice. Histologically, *Slc40a1 C326Y^fl/fl;Myh6.Cre+* hearts from six month old mice also showed LV enlargement (**Figure 3D**), hypertrophied cardiomyocytes (**Figure 3E–F**), upregulation of hypertrophy markers *Myh7* and *Nppb* (**Figure 3G**) and a greater degree of apoptosis compared to *Slc40a1 C326Y^fl/fl* controls (**Figure 3H–I**). When we measured cardiac performance by cine MRI, we found that *Slc40a1 C326Y^fl/fl;Myh6.Cre+* mice also developed LV dysfunction by 6 months of age, with a reduction in LVEF from 73% to 55% (**Figure 3J–L**).

The similarity between *Hamp^fl/fl;Myh6.Cre+* and *Slc40a1 C326Y^fl/fl;Myh6.Cre+* mice in terms of the nature and time course of cardiac dysfunction suggests a common mechanism of cardiac dysfunction involving upregulation of cardiomyocyte FPN. Therefore, we tested whether this upregulation of FPN resulted in increased iron efflux from cardiomyocytes. Iron Fe55 efflux was indeed significantly greater in cardiomyocytes isolated from *Hamp^fl/fl;Myh6.Cre+* and *Slc40a1 C326Y^fl/fl; Myh6.Cre+* hearts than in cardiomyocytes isolated from their respective controls (**Figure 3M**). Addition of exogenous mouse HAMP in the efflux medium inhibited the increase in Fe55 efflux from *Hamp^fl/fl;Myh6.Cre+* cardiomyocytes but not from *Slc40a1 C326Y^fl/fl;Myh6.Cre+* cardiomyocytes, consistent with the HAMP-resistant mutation in *Slc40a1 C326Y^fl/fl;Myh6.Cre+* cardiomyocytes. We hypothesised that upregulation of cardiac FPN and iron export in *Hamp^fl/fl;Myh6.Cre+* and *Slc40a1 C326Y^fl/fl;Myh6.Cre+* hearts caused cardiomyocyte iron depletion. To test this hypothesis, we quantified iron levels both in total hearts and in the isolated cardiomyocyte fractions at 3 months and 6 months of age. While iron levels in total hearts were not significantly different between any of the genotypes (**Figure 3—figure supplement 2**), the iron content of the cardiomyocyte fraction was significantly lower in *Hamp^fl/fl;Myh6.Cre+* and in *Slc40a1 C326Y^fl/fl;Myh6.Cre+* mice than in their respective controls (**Figure 3N**). Furthermore, expression of TfR1 mRNA was upregulated (**Figure 3O**) and Slc40a1 mRNA was downregulated (**Figure 3P**) in *Hamp^fl/fl;Myh6.Cre+* and *Slc40a1 C326Y^fl/fl;Myh6.Cre+* hearts relative to their respective controls, consistent with a transcriptional response to intracellular iron deficiency (**Rouault and Klausner, 1997**; **Rouault, 2006**; **Ward and Kaplan, 1823**). Together these results demonstrate that loss of either cardiac hepcidin or hepcidin responsiveness in the heart results in upregulation of cardiomyocyte FPN, and that cardiomyocytes of *Hamp^fl/fl;Myh6.Cre+* and *Slc40a1 C326Y^fl/fl;Myh6.Cre+* hearts are iron deficient as a result of upregulation of FPN-mediated iron export.

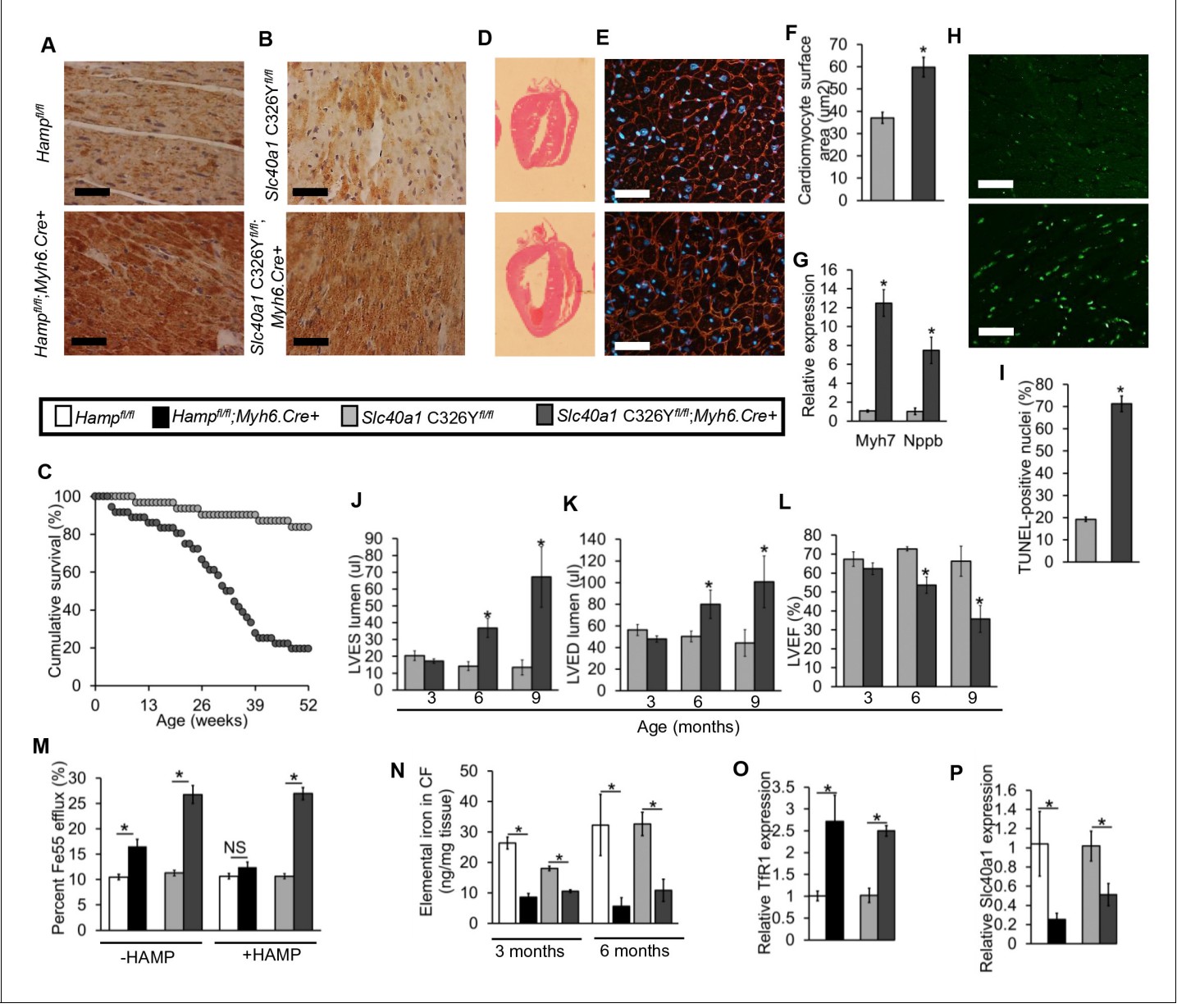

**Figure 3.** The role of cardiomyocyte FPN. (A–B) Immunohistochemical staining for FPN in the hearts of three month old *Hamp^fl/fl^;Myh6.Cre+*, *Slc40a1 C326Y^fl/fl^;Myh6.Cre+* mice and respective controls. (C) Cumulative survival of *Slc40a1 C326Y^fl/fl^;Myh6.Cre+* mice (n = 36) and *Slc40a1 C326Y^fl/fl^* littermate controls (n = 31) over 52 weeks. (D) Representative H and E-stained longitudinal heart sections from a six month old *Slc40a1 C326Y^fl/fl^;Myh6.Cre+* mouse and a *Slc40a1 C326Y^fl/fl^* control. (E) Representative WGA cardiac staining from a six month old *Slc40a1 C326Y^fl/fl^;Myh6.Cre+* mouse and *Slc40a1 C326Y^fl/fl^* control. (F) Quantitation of cardiomyocyte size based on WGA staining. n = 3 per group. *p=0.001 relative to *Slc40a1 C326Y^fl/fl^* controls. (G) Relative expression of *Myh7* and *Nppb* in hearts of 6 month old *Slc40a1 C326Y^fl/fl^;Myh6.Cre+* mice and *Slc40a1 C326Y^fl/fl^* controls. n = 3 per group. *p=0.032, 0.044 for the respective gene relative to *Slc40a1 C326Y^fl/fl^* controls. (H) Representative cardiac TUNEL staining from a six month old *Slc40a1 C326Y^fl/fl^;Myh6.Cre+* mouse and *Slc40a1 C326Y^fl/fl^* control. (I) Quantitation of percentage of apoptotic cardiomyocytes based on TUNEL staining, n = 3 per group. *p=0.0003 relative to *Slc40a1 C326Y^fl/fl^* controls. (J–L) Cine MRI measurements of LVES, LVED and LVEF in *Slc40a1 C326Y^fl/fl^;Myh6.Cre+* mice and *Slc40a1 C326Y^fl/fl^* controls at three months (n = 6 per group), six months (n = 6 per group, *p=0.003 for LVES, 0.043 for LVED and 0.001 for LVEF) and nine months (n = 5 per group, *p=0.033 for LVES, 0.047 for LVED and 0.023 for LVEF). P values are relative to *Slc40a1 C326Y^fl/fl^* controls of the same age. (M) Percentage Fe55 efflux in cardiomyocytes from *Hamp^fl/fl^;Myh6.Cre+* mice, *Slc40a1 C326Y^fl/fl^;Myh6.Cre+* mice and respective controls, in presence or absence of HAMP peptide. *p=0.018, 0.006 and 0.007 respectively (N) Elemental iron levels in cardiomyocyte fractions (CF) from the hearts of *Hamp^fl/fl^;Myh6.Cre+* mice, *Slc40a1 C326Y^fl/fl^;Myh6.Cre+* mice and their respective controls. n = 4 per group. *p=0.032, 0.044, 0.047 and 0.031 respectively. (O–P) Relative *TfR1* (*p=0.038, 0.001) and *Fpn* (*p=0.039, 0.047) expression in hearts of 3 month old *Hamp^fl/fl^;Myh6.Cre+* mice, *Slc40a1 C326Y^fl/fl^;Myh6.Cre+* mice and their respective controls. All values are plotted as mean ± SEM. Scale bar = 50 µm.

*Figure 3 continued on next page*

*Figure 3 continued*

The following source data and figure supplements are available for figure 3:

**Source data 1.** Source data file for *Figure 3JK and L*.
**Figure supplement 1.** Characterisation of *Slc40a1* C326Y[fl/fl];*Myh6.Cre+* mice.
**Figure supplement 2.** Elemental iron levels in total hearts of *Hamp[fl/fl]*;*Myh6*.

## The role of cardiomyocyte iron deficiency and metabolic derangement in cardiac dysfunction

As cardiomyocyte iron deficiency preceded the development of cardiac dysfunction, we hypothesised it is the cause of the cardiac phenotype in *Hamp[fl/fl]*;*Myh6.Cre+* mice. To test this hypothesis, we treated *Hamp[fl/fl]*;*Myh6.Cre+* mice and *Hamp[fl/fl]* controls with fortnightly intravenous injections of ferric carboxymaltose solution containing 0.5 mg iron from three months of age, and confirmed the effects of this treatment on cardiac and systemic iron indices at 6 months of age (*Table 4*). At this timepoint, we performed cine MRI and found that the LV enlargement and the reduced LVEF, seen in untreated *Hamp[fl/fl]*;*Myh6.Cre+* mice, were prevented in iron-treated *Hamp[fl/fl]*;*Myh6.Cre+* mice (*Figure 4A–C*). The transcriptional response to intracellular iron deficiency in untreated *Hamp[fl/fl]*; *Myh6.Cre+* mice (upregulation of TfR1 mRNA and downregulation of Slc40a1 mRNA relative to *Hamp[fl/fl]* controls), was absent in iron-treated *Hamp[fl/fl]*;*Myh6.Cre+* mice, consistent with correction of cardiomyocyte iron deficiency (*Figure 4D–E*). Prevention of cardiac dysfunction in *Hamp[fl/fl]*;*Myh6.Cre+* mice by intravenous iron treatment confirms the causal relationship between cardiomyocyte iron deficiency and cardiac dysfunction in this setting.

Having confirmed a causal relationship between cardiomyocyte iron deficiency and cardiac dysfunction, we aimed to understand the mechanisms linking the two. Iron is a cofactor for several enzymes involved in metabolism (*Meyer, 2008*; *Sono et al., 1996*; *Solomon et al., 2003*), and metabolic derangement is a well-recognized precursor to cardiac dysfunction (*Belke et al., 2000*; *Kakinuma et al., 2000*; *Ashrafian et al., 2007*). Iron deficiency has been reported to reduce the levels and/or activities of key metabolic iron-containing enzymes, in cell lines, in the hearts of mice with impaired cardiac iron uptake (cardiac-specific *TfR1* knockouts) and in the hearts of mice fed an iron-deficient diet (*Dallman, 1986*; *Dhur et al., 1989*; *Xu et al., 2015*; *Oexle et al., 1999*). Based on those studies, we postulated that iron deficiency in the cardiomyocytes of *Hamp[fl/fl]*;*Myh6.Cre+* mice would also result in reduction in the activities of key metabolic iron-containing enzymes. To test this hypothesis, we measured the activities of the iron-sulphur containing enzyme Aconitase I as well as electron transport chain (ETC) complexes in cardiac lysates from *Hamp[fl/fl]*;*Myh6.Cre+* hearts at 3 months and 6 months of age. We found that Aconitase I, Complex I and Complex IV activities were significantly reduced in *Hamp[fl/fl]*;*Myh6.Cre+* hearts compared to *Hamp[fl/fl]* controls at 3 months and 6 months of age, and that this reduction in activity was prevented in iron-treated 6-month old mice

**Table 3.** Characterisation of *Slc40a1* C326Y[fl/fl];*Myh6.Cre+* mice.
Indices of iron status in *Slc40a1* C326Y[fl/fl] and *Slc40a1* C326Y[fl/fl];*Myh6.Cre+* mice at six months of age (n = 4 per group). Values are shown as mean ± SEM.

|  | *Slc40a1* C326Y[fl/fl] | *Slc40a1* C326Y [fl/fl]; *Myh6.Cre+* |
|---|---|---|
| liver total elemental iron (ng/mg tissue) | 92.77 ± 21.30 | 84.00 ± 26.00 |
| liver ferritin (μg/mg total protein) | 0.87 ± 0.06 | 0.92 ± 0.05 |
| serum iron (μmol/L) | 27.30 ± 5.20 | 29.60 ± 7.20 |
| serum ferritin (mg/L) | 2.10 ± 0.04 | 2.20 ± 0.15 |
| hemoglobin (g/L) | 125.70 ± 8.80 | 126.00 ± 12.30 |
| serum hepcidin (μg/L) | 25.90 ± 11.60 | 27.50 ± 8.40 |

**Table 4.** Effect of intravenous iron treatment on iron indices.

Total cardiac and liver elemental iron, serum iron and circulating HAMP in 6-month old untreated and I.V iron-treated $Hamp^{fl/fl}$;Myh6. Cre+ mice and $Hamp^{fl/fl}$ littermate controls. Treated mice were injected with 0.5 mg iron fortnightly from the age of 3 months. Tissues and serum were harvested 12 hr after the final injection. n = 5 per group. *p<0.05 relative to untreated $Hamp^{fl/fl}$ mice. †p < 0.05 relative to untreated $Hamp^{fl/fl}$;Myh6.Cre+. Values are shown as mean ± SEM.

| | $Hamp^{fl/fl}$ | | $Hamp^{fl/fl}$;Myh6.Cre+ | |
| --- | --- | --- | --- | --- |
| | untreated | treated with I.V iron | untreated | treated with I.V iron |
| cardiac total elemental iron (ng/mg tissue) | 82.2 ± 16.9 | 331.3 ± 21.5* | 74.9 ± 7 | 399.8 ± 68.5† |
| liver total elemental iron (ng/mg tissue) | 100.4 ± 11 | 2527.6 ± 27.63* | 96.3 ± 14 | 2258.2 ± 239.9† |
| serum iron (μmol/L) | 30.09 ± 6.37 | 74.48 ± 17.96* | 31.5 ± 6.9 | 80.12 ± 24.9† |
| serum hepcidin (μg/L) | 27.41 ± 6.7 | 237.3 ± 16.7* | 28.9 ± 9.4 | 209.8 ± 38.8† |

(**Figure 4F–H**). As ETC activity is essential to mitochondrial function, we examined whether $Hamp^{fl/fl}$; Myh6.Cre+ hearts had signs of mitochondrial failure, and whether such mitochondrial failure was prevented by intravenous iron supplementation. By electron microscopy (EM), dilation of mitochondrial cristae was seen in $Hamp^{fl/fl}$;Myh6.Cre+ hearts as early as 3 months of age and progressed further at 6 months of age. However, this was prevented in iron-treated $Hamp^{fl/fl}$;Myh6.Cre+ mice, which had healthy-looking mitochondria at 6 months of age (**Figure 4I**). Similar changes in mitochondrial morphology were seen in the hearts of 3 month old $Slc40a1 C326Y^{fl/fl}$;Myh6.Cre+ mice (data not shown).

Impairment of electron transport is known to drive glycolysis, as an alternative route of ATP production (**Schönekess et al., 1997**; **Kato et al., 2010**). Therefore, we examined the expression of a number of genes encoding glycolytic enzymes. We found that expression of genes encoding Hexokinase 2 (Hk2) which catalyses the first step of glycolysis, Enolase (Eno), which catalyses the penultimate step of glycolysis and of Lactate dehydrogenase A (Ldha) which catalyses the ultimate step of glycolysis were all significantly increased at 3 months and 6 months of age in hearts of untreated $Hamp^{fl/fl}$;Myh6.Cre+ mice. This upregulation of glycolytic genes was not seen in iron-treated $Hamp^{fl/fl}$;Myh6.Cre+ mice (**Figure 4J–L**). These results demonstrate that reduction in the activities of key iron-containing metabolic enzymes, mitochondrial failure and upregulated glycolysis precede the development of cardiac dysfunction in $Hamp^{fl/fl}$;Myh6.Cre+ hearts, and are prevented by intravenous iron treatment.

## Discussion

The major finding of this study is that cardiomyocyte hepcidin is required for autonomous cellular iron homeostasis. Loss of hepcidin responsiveness specifically in cardiomyocytes engendered the same effects as loss of cardiac hepcidin, demonstrating that cardiac hepcidin operates in an autocrine fashion by regulating cardiomyocyte FPN. A role in cellular iron homeostasis, and an autocrine mode of action for the HAMP/FPN axis have not been described previously in any other tissue. Indeed, hepcidin and FPN are better known to interact in an endocrine fashion, at the level of the gut, spleen and liver, to regulate systemic iron homeostasis.

A second important finding is that this cardiac HAMP/FPN axis is essential for normal heart function, and that its disruption leads to ultimately fatal cardiac metabolic and contractile dysfunction, even against a background of intact systemic iron homeostasis. Metabolic and contractile dysfunction are preceded by cardiomyocyte iron deficiency and prevented by intravenous iron supplementation, indicating a causal relationship between cardiomyocyte iron deficiency and cardiac dysfunction. This causal relationship has previously been demonstrated in studies using mice lacking cardiac TfR1, in which cardiac iron deficiency also affects the heart against a background of otherwise intact systemic iron homeostasis (**Xu et al., 2015**). Furthermore, the importance of metabolic derangement seen in our mouse models is also consistent with the findings in mice lacking the cardiacTfR1, and in mice and rats fed iron deficient diets (**Dhur et al., 1989**; **Xu et al., 2015**; **Tanne et al., 1994**; **Walter et al., 2002**). Correction of metabolic and contractile dysfunction by intravenous iron

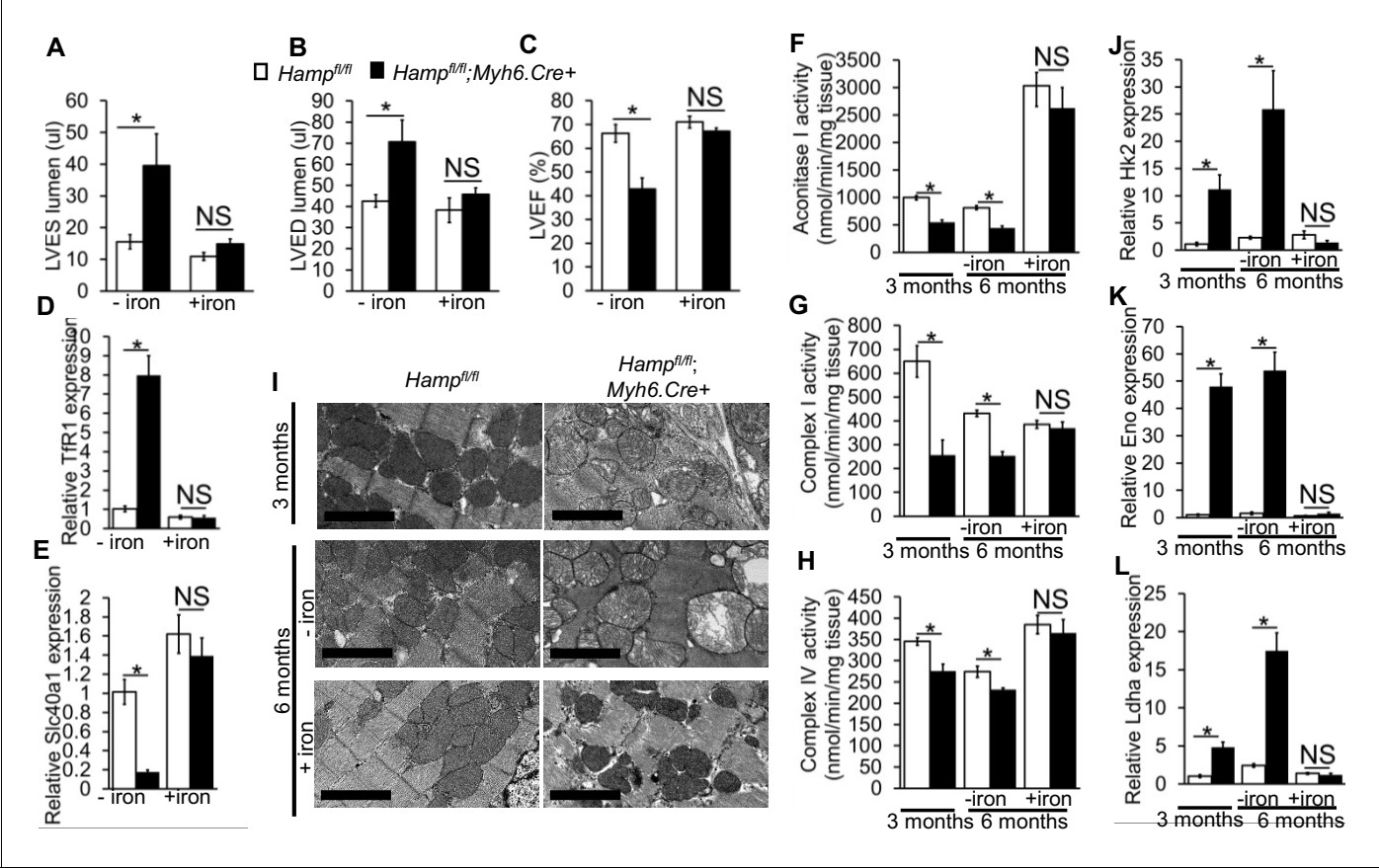

**Figure 4.** The role of cardiomyocyte iron deficiency and metabolic derangement in cardiac dysfunction. (A–C) Cine MRI measurements of LV lumen, at end-systole (LVES, *p=0.048), end-diastole (LVED, *p=0.031), and of ejection fraction (LVEF, *p=0.004) in 6-month old untreated (-iron) and I.V iron-treated (+iron) *Hamp<sup>fl/fl</sup>;Myh6.Cre+* mice and *Hamp<sup>fl/fl</sup>* littermate controls. n = 5 per group. (D–E) Relative *TfR1* (*p=0.001) and *Fpn* (*p=0.002) expression in hearts of 6-month old untreated (-iron) and I.V iron-treated (+iron) *Hamp<sup>fl/fl</sup>;Myh6.Cre+* mice and *Hamp<sup>fl/fl</sup>* littermate controls. n = 4 per group. (F–H) Enzymatic activities of Aconitase I (*p=0.035, 0.041), Complex I (*p=0.004, 0.030) and Complex IV (*p=0.003, 0.026) in untreated (-iron) 3-month and 6-month old and in I.V iron-treated (+iron) 6-month old *Hamp<sup>fl/fl</sup>;Myh6.Cre+* mice and *Hamp<sup>fl/fl</sup>* littermate controls. n = 4 per group. (I) Representative EM micrographs of hearts from untreated (-iron) 3-month and 6-month old and I.V iron-treated (+iron) 6-month old *Hamp<sup>fl/fl</sup>;Myh6.Cre+* mice and *Hamp<sup>fl/fl</sup>* littermate controls. Scale bar = 2 µm. (J–L) Relative *Hk2* (*p=0.010, 0.002), *Eno* (*p=0.014, 0.021) and *Ldha* (*p=0.003, 0.001) expression levels in hearts of untreated (-iron) 3-month and 6-month old and in I.V iron-treated (+iron) 6-month old *Hamp<sup>fl/fl</sup>;Myh6.Cre+* mice and *Hamp<sup>fl/fl</sup>* littermate controls. n = 4 per group. NS=not significant. All values are plotted as mean ± SEM.

The following source data is available for figure 4:

**Source data 1.** Source data file for *Figure 4A,B and C*.

treatment likely involves not only increased iron availability for uptake into cardiomyocytes, but also the effects of increased circulating HAMP on cardiomyocyte FPN.

Surprisingly, we found that, both in vitro and in vivo, cardiac HAMP protein responded differently from its transcript to changes in iron levels. The current understanding of hepcidin regulation is based on studies of hepatic hepcidin. In that setting, release of the active mature HAMP peptide is dependent on cleavage of the propeptide by Furin (*Valore and Ganz, 2008*). We found that Furin inhibition increased iron export from *Hamp<sup>fl/fl</sup>* but not from *Hamp<sup>fl/fl</sup>;Myh6.Cre+* cardiomyocytes (*Appendix 1—figure 1*), demonstrating that cardiomyocytes secrete an active HAMP peptide in a Furin-dependent manner. Furthermore, we found that increased HAMP release from iron-deficient cardiomyocytes depended on Furin, and that cardiac Furin itself is upregulated by iron deficiency both in vitro and in vivo. The latter finding is consistent with the reported regulation of Furin by iron deficiency through Hypoxia-Inducible Factors HIFs (*Silvestri et al., 2008*). These data suggest that

differential regulation of Furin by iron may explain the divergent effects of iron on Hamp transcript and HAMP protein. Comprehensive studies using cardiac-specific knockouts of putative regulators will be required to explore formally the regulation or otherwise, of cardiac hepcidin by pathways known to regulate hepatic hepcidin.

The upregulation of cardiac HAMP in mice fed an iron-deficient diet raises the possibility that it may be involved in protecting the heart in the setting of systemic iron deficiency. This hypothesis is supported by the finding that, when provided an iron-deficient diet, *Hamp^{fl/fl};Myh6.Cre+* mice exhibited a greater cardiac hypertrophic response than their *Hamp^{fl/fl}* littermate controls (**Appendix 1—figure 2**).

Intracellular iron levels are dependent both on cellular homeostatic pathways and on systemic iron availability in plasma. Therefore, the interplay between the cardiac and the systemic HAMP/FPN axes is important in determining cardiomyocyte iron levels. Some insight into this interplay is gained from comparing systemic and cardiac mouse models of disrupted iron homeostasis. It is interesting that ubiquitous *Hamp* knockout (**Lakhal-Littleton et al., 2015**) and ubiquitous *Slc40a1 C326Y* knock-in mice (**Appendix 1—figure 3**), both models of systemic iron overload, do not develop the cardiac dysfunction seen in cardiomyocyte-specific *Hamp* knockout and cardiomyocyte-specific *Slc40a1 C326Y* knock-in mice. This suggests that, while upregulation of cardiomyocyte FPN under conditions of normal iron availability (cardiac-specific models described in this study) is detrimental to cardiac function, it is protective under conditions of increased systemic iron availability (systemic models). Previously, we also showed that deletion of cardiomyocyte FPN resulted in fatal cardiomyocyte iron overload, preventable by dietary iron restriction (**Lakhal-Littleton et al., 2015**). Together, our studies demonstrate that iron levels within cardiomyocytes are a balance between cellular iron efflux which is regulated by the cardiac HAMP/FPN axis, and systemic iron availability which is regulated by the systemic HAMP/FPN axis (**Figure 5**).

Iron overload is detrimental to cardiac health, as demonstrated by iron overload cardiomyopathy in hemochromatosis and thalassemia major patients (**Gulati et al., 2014**). Our model of cardiac iron homeostasis implies that the cardiac HAMP/FPN axis may have a modifying effect on the severity of iron-overload cardiomyopathy. Thus, it would be interesting to explore whether differences in the levels of cardiac FPN and HAMP, possibly due to different local stimulatory and suppressive signals (e.g local inflammation, local ischemia), explain the reported lack of concordance between the degrees of cardiac iron overload and liver iron overload in a significant proportion of hemochromatosis and thalassemia major patients (**Anderson et al., 2001**; **Noetzli et al., 2008**).

Iron deficiency is also detrimental to cardiac health. Indeed, systemic iron deficiency correlates with functional and molecular markers of disease severity in patients with chronic heart failure (CHF) (**Klip et al., 2013**; **Comín-Colet et al., 2013**),and also appears to contribute to the risk of death after an episode of acute heart failure (**Jankowska et al., 2014**). Given the high prevalence of iron deficiency in patients with CHF, ranging between 30–50% (**Klip et al., 2013**; **Nanas et al., 2006**; **Ezekowitz et al., 2003**), the European Society of Cardiology recently recommended the assessment of iron deficiency as a comorbidity in CHF. Furthermore, several clinical trials have now established the benefits of intravenous iron supplementation in CHF patients, with or without anaemia (**Ponikowski et al., 2015**; **Anker et al., 2009**; **Avni et al., 2012**). The mechanisms underlying the anaemia-independent effects of iron deficiency and the benefits of intravenous iron in CHF patients are not fully understood. In light of the direct effect of cardiomyocyte iron deficiency on heart function, demonstrated in this and other studies, it would interesting to explore whether systemic iron deficiency in CHF patients is accompanied by cardiomyocyte iron deficiency, and whether correction of the later underlies the benefits of intravenous iron supplementation in non-anaemic patients.

Another open question is whether disruption of the cardiac HAMP/FPN axis contributes to the pathophysiology of heart disease. It has been shown, in the rat model of myocardial infarction (MI), that Hamp mRNA and HAMP protein are elevated in the ischemic myocardium during the acute phase (**Simonis et al., 2010**). In humans, circulating HAMP was shown to be elevated in the serum within 4 hr of MI, although the tissue source of this hepcidin was not identified (**Suzuki et al., 2009**). In addition, decreased cardiac HAMP expression has been reported in a transgenic mouse model of dilated cardiomyopathy, where the phenotype was ameliorated following transgenic overexpression of cardiac hepcidin (**Zhang et al., 2012**). Thus, further studies are warranted in humans to explore formally the role of the cardiac HAMP/FPN axis in the aetiology of heart disease.

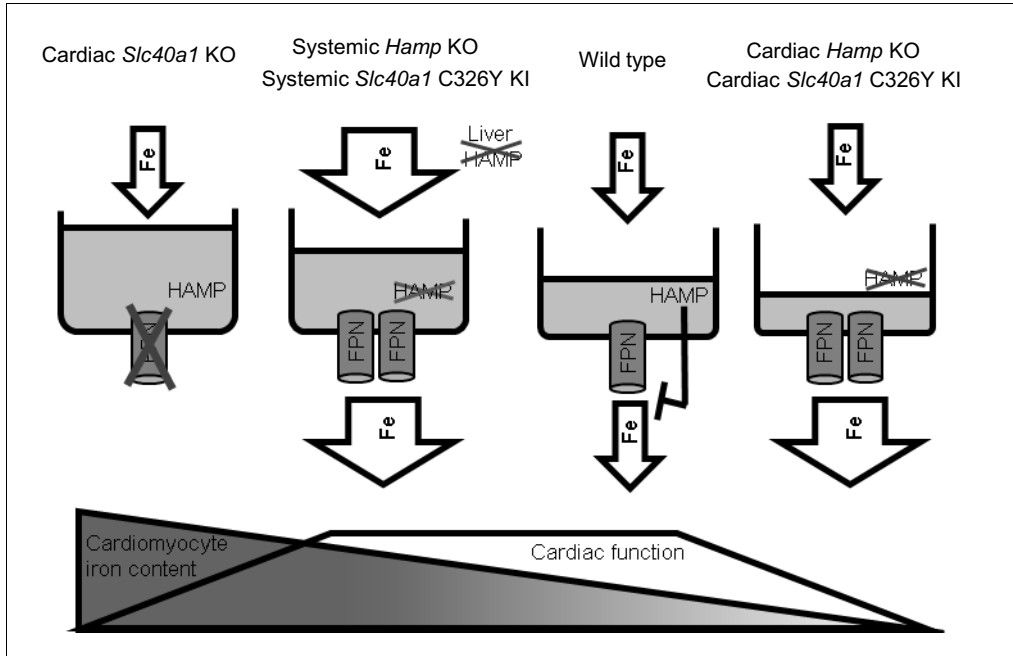

**Figure 5.** Interplay between systemic and cardiac iron HAMP/FPN axes. Cardiomyocyte iron content is determined by both systemic iron availability, which is regulated by liver HAMP, and by the cardiac HAMP/FPN axis, which regulates cardiomyocyte iron efflux. In the wild type heart, cardiac HAMP regulates the levels of cardiac FPN and iron release from cardiomyocytes. In this study, we have demonstrated that loss of cardiac HAMP (cardiac Hamp KO) or loss of cardiac HAMP responsiveness (cardiac Slc40a1 C326Y KI) result in cardiomyocyte iron deficiency due to increased cardiomyocyte FPN and iron release. Previously, we also demonstrated that loss of cardiomyocyte FPN caused cardiomyocyte iron overload. In these two sets of conditions, cardiomyocyte iron deficiency and cardiomyocyte iron overload cause cardiac dysfunction. We have also shown that upregulation of cardiac FPN occurs as a result of loss of either systemic HAMP or systemic HAMP responsiveness, and is protective against the otherwise detrimental effects of systemic iron overload.

Currently, there is considerable interest in targeting the HAMP/FPN axis for the treatment of iron overload and iron deficiency. Our studies suggest that such strategies may also impinge on cardiac iron homeostasis and function. Other than the heart, both FPN and hepcidin are also found in the brain, kidney and placenta (*McCarthy and Kosman, 2014*; *Kulaksiz, 2005*; *Evans et al., 2011*; *Rouault, 2013*; *Moos and Rosengren Nielsen, 2006*; *Bastin et al., 2006*; *Wolff et al., 2011*). It would be important to establish whether our findings in the heart extend to these tissues.

## Materials and methods

### Mice

All animal procedures were compliant with and approved under the UK Home Office Animals (Scientific Procedures) Act 1986. Both males and females were used in experiments, with the respective littermate control being of a matching sex.

The strategy for generating cardiac Hamp knockout mice is outlined in *Appendix 1—figure 4*. Briefly, a targeting vector was designed to introduce a floxed *Hamp* allele into C57BL/6N mouse ES cells (JM8F6) with exons 2 and 3, which encode the majority of the peptide, flanked by LoxP sites and a line of floxed mice was generated by blastocyst injection of targeted ES cells, as previous described (*Lakhal-Littleton et al., 2015*). Further breeding with a C57BL/6 Flp recombinase deleter mouse allowed removal of the Neomycin resistance cassette. Cardiac *Hamp* knockouts were then generated by by crossing *Hamp*fl/fl animals with mice transgenic for Cre recombinase, under the control of cardiomyocyte-specific Alpha Myosin Heavy Chain (*Myh6*) promoter (B6.FVB-Tg(*Myh6*-cre)

2182Mds/J).The subsequent breeding strategy was designed to produce cardiac *Hamp* knockouts and homozygous floxed controls (*Hamp^fl/fl^;Myh6.Cre+* and *Hamp^fl/fl^* respectively) in the same litter.

The strategy for generating Cardiac *Slc40a1 C326Y* knock-in mice is outlined in *Appendix 1—figure 5*, and further details are provided in the Appendix.

## Cine MRI

Mice were anaesthetized with 2% isofluorane in $O_2$ and positioned supine in a purpose-built cradle. ECG electrodes were inserted into the forepaws, a respiration loop was taped across the chest and heart and respiration signals were monitored using a custom-built physiological motion gating device. The cradle was lowered into a vertical-bore, 11.7 T MR system with a 40 mm birdcage coil (Rapid Biomedical, Würzburg, Germany) and visualised using a Bruker console running Paravision 2.1.1. A stack of contiguous 1 mm thick true short-axis ECG and respiration-gated cine-FLASH images were acquired. The entire in vivo imaging protocol was performed in approximately 60 min. Image analysis was performed using ImageJ (NIH Image, Bethesda, MD). Left ventricular volumes and ejection fractions were calculated from the stack of cine images as described (*Lakhal-Littleton et al., 2015*).

## Dietary iron content

Unless otherwise stated, animals were provided with a standard rodent chow diet containing 200 ppm iron. In iron manipulation experiments, mice were given an iron-deficient diet (2–5 ppm iron; Teklad TD.99397; Harlan Laboratories), or an iron-loaded diet (5000 ppm iron; Teklad TD.140464) or a matched control diet (200 ppm iron; Teklad TD.08713) from weaning for six weeks.

## Isolation of primary adult mouse cardiomyocytes and in vitro treatment

Adult primary cardiomyocytes were isolated from eight week old C57BL/6 mice. Hearts were cannulated and mounted on a langendorff apparatus, then perfused using a liberase solution for 10 min. After filtration through a 400 µm gauze, cells were cultured in MEM medium containing Hanks salts, L-glutamine and antibiotics. Within 2 hr of cardiomyocyte culture, supernatants were replaced with fresh medium containing 10% Fetal calf serum, with 0.5 mmol/L ferric citrate (FAC) (F3388, Sigma Aldrich) or 100 µmol/L desferroxamine (D9533, Sigma Aldrich) for 8 hr. The Furin inhibitor decanoyl-Arg-Val-Lys-Arg-chloromethylketone (CMK) (N1505, Bachem) was added at a concentration of 50 µmol/L for the duration of DFO and FAC treatment.

## Quantitative PCR

Total RNA extraction and cDNA synthesis were carried out as previously described (*Lakhal-Littleton et al., 2015*). Gene expression was measured using Applied Biosystems Taqman gene expression assay probes for *Slc40a1, Hamp, TfR1, Myh7, Nppb, Ldha, Hk2, Eno* and house-keeping gene *β-Actin* (Life Technologies, Carlsbad, CA). The CT value for the gene of interest was first normalised by deducting CT value for *β-Actin* to obtain a delta CT value. Delta CT values of test samples were further normalised to the average of the delta CT values for control samples to obtain delta delta CT values. Relative gene expression levels were then calculated as $2^{-delta\ deltaCT}$.

## Immunohistochemistry

Tissues were prepared as described previously (*Lakhal-Littleton et al., 2015*) and stained with rabbit polyclonal anti-mouse HAMP antibody (ab30760, Abcam, RRID:AB_2115844) at 1/40 dilution, or rabbit polyclonal anti-mouse FPN antibody (MTP11-A, Alpha Diagnostics, RRID:AB_1619475) at 1/200 dilution. Results of control experiments confirming the specificity of the HAMP antibody are shown in *Figure 1—figure supplement 5*.

## HAMP enzyme-linked immunosorbent assay (ELISA)

HAMP was measured in mouse sera and in cardiomyocyte supernatants using a HAMP ELISA kit (E91979Mu, USCN) according to the manufacturer's instructions. Results of control experiments confirming that in vitro treatments did not affect HAMP peptide detection by this ELISA kit are shown in *Figure 1—figure supplement 6*.

## DAB-enhanced perls stain

Formalin-fixed paraffin-embedded tissue sections were deparaffinised using Xylene, then rehydrated in ethanol. Slides were then stained for 1 hr with 1% potassium ferricyanide in 0.1 mol/L HCl buffer. Endogenous peroxidase activity was quenched, then slides were stained with DAB chromogen substrate and counterstained with haematoxylin. They were visualised using a standard brightfield microscope.

## Electron microscopy (EM)

Hearts were dissected and 0.5–1 mm$^3$ slices were fixed by immersion for 2 hr in 2.5% glutaraldehyde in 0.1 mol/L cacodylate buffer and prepared for electron microscopy by standard methods. Briefly, cells were post-fixed in osmium tetroxide (1% w/v in 0.1 mol/L phosphate buffer), stained with uranyl acetate (2% w/v in distilled water), dehydrated through increasing concentrations of ethanol (70–100%) and acetone and embedded in TAAB resin (TAAB, Aldermaston, UK). Ultrathin sections (50–80 nm) were prepared using a Reichert ultracut S microtome and mounted on 200 mesh nickel grids. Sections were lightly counterstained with lead citrate and uranyl acetate and examined with a Jeol transmission electron microscope (JEM-1010, JEOL, Peabody MA).

## Fe55 efflux in primary adult cardiomyocytes

Adult cardiomyocytes were isolated from mice of the desired genotype at 9 weeks of age as described above. Cardiomyocytes were then cultured in 24-well plates at equal densities for 16 hr before the efflux experiment was performed as described (*McKie et al., 2000*). Briefly, after washing with three times PBS, cells were incubated for 30 min in 200 µl uptake solution (98 mmol/L NaCl, 2.0 mmol/L KCl, 0.6 mmol/L CaCl2, 1.0 mmol/L MgCl2, 1.0 mmol/L ascorbic acid, 10 mmol/L HEPES [pH 6.0] with Tris base, 50 µmol/L Fe55 [NEN, Boston, MA]), then washed three times with PBS and incubated for 30 min with efflux solution (98 mmol/L NaCl, 2.0 mmol/L KCl, 0.6 mmol/L CaCl2, 1.0 mmol/L MgCl2, 10 mmol/L HEPES [pH 7.4] with Tris base, 300 U/ml bovine ceruloplasmin (cp) (Sigma) and 40 µg/ml human apotransferrin (tf) [Sigma]),in the absence or presence of 0.5 µmol/L mouse HAMP peptide (Peptides International). The efflux medium was then removed, the cells washed three times in ice-cold PBS and disrupted by incubation in 100 µl of 10% SDS solution for 10 min.The efflux solution and cell lysates were then transferred into scintillation vials for Fe55 counting. Where Furin inhibition was carried out, CMK was added to the culture medium at 50 µmol/L 2 hr before the efflux experiment was carried out.

## Iron quantitation

Ferritin concentration in serum and in liver lysates was determined using the ferritin ELISA kit (ICL, Inc. Portland). Serum iron levels were determined using the ABX-Pentra system (Horiba Medical, CA). Determination of total elemental iron in the heart was carried out by inductively coupled plasma mass spectrometry (ICP-MS) as described previously (*Lakhal-Littleton et al., 2015*). Calibration was achieved using the process of standard additions, where spikes of 0 ng/g, 0,5 ng/g, 1 ng/g, 10 ng/g, 20 ng/g and 100 ng/g iron were added to replicates of a selected sample. An external iron standard (High Purity Standards ICP-MS-68-A solution) was diluted and measured to confirm the validity of the calibration. Rhodium was also spiked onto each blank, standard and sample as an internal standard at a concentration of 1 ng/g. Concentrations from ICP-MS were normalised to starting tissue weight.

## Isolation of cardiomyocyte and non-cardiomyocyte fractions for iron quantitation

Following cardiac perfusion, hearts were dissected into small pieces in ice-cold Hanks buffer, and subject to collagenase P digestion at 37C for 1 hr (11213857001, Roche Diagnostics). Following lysis of red blood cells, cell suspensions were passed through a 70 µm sieve, before being labelled using cardiomyocyte isolation kit (130-100-825, Miltenyi Biotec). Separation of cardiomyocyte and non-cardiomyocyte fractions was carried out using MACS magnetic separation system according to the manufacturer's instructions. Cardiac fractions were lysed immediately for ICP-MS analysis.

## Activity assays for aconitase I and electron transport chain (ETC) complexes

Approximately 10 mg of frozen, crushed tissues were suspended in 200 μl of ice-cold KME buffer (100 mmol/L KCl, 50 mmol/L Mops, 0.5 mmol/L EGTA, pH 7.4), then homogenized by rupturing with a TissueRuptor (Qiagen, UK) over ice. In a plastic cuvette, the cardiac lysate is mixed with assay buffer and slotted into a spectrophotometer. Details of assay buffers and of reaction procedure for each enzyme are detailed in the Appendix.

## Statistics

Values are shown as mean ± standard error of the mean (SEM). Comparison of iron indices, enzyme activities and parameters of cardiac function between groups was performed using Student's T test. $p$ values $< 0.05$ were deemed as indicating significant differences between groups. Where significant, exact $p$ values for a figure panel are stated in the corresponding figure legend. No explicit power analysis was performed prior to the experiments to determine sample size, since we had no means to reliably estimate the size and variability of the effects of deleting hepcidin on parameters of cardiac function. For Cine MRI assessment of cardiac function, typically 5–11 animals of each genotype were used, with a matching number of littermate controls. For gene expression, iron quantitation and enzyme activity assays from mouse tissues, typically, 3–6 independent biological replicates and matching littermate controls were analysed. Since significant results were obtained from these set of experiments, no further animals were sacrificed. All 'n' values reported refer to independent biological replicates.

## Acknowledgements

We thank Yixing Wu, MRC Functional Genomic Unit, University of Oxford for his help with preparation of reagents for ETC complex assays.

We thank Prof Keith Dorrington, Dr Matthew Frise (Department of Physiology, Anatomy and Genetics, University of Oxford) and Dr Edward Littleton (University Hospitals Birmingham NHS Foundation Trust, Queen Elizabeth Hospital), for critical reading of the paper.

## Additional information

### Funding

| Funder | Grant reference number | Author |
|---|---|---|
| British Heart Foundation | FS/12/63/29895 | Samira Lakhal-Littleton |
| Wellcome Trust | 090532/Z/09/Z | Samira Lakhal-Littleton |
| Vifor Pharma | R19874-CN004 | Magda Wolna |

The funders funded the salary and research expenses relating to this work for two of the authors, as described above.

### Author contributions

SL-L, Conception and design, Acquisition of data, Analysis and interpretation of data, Drafting or revising the article; MW, Acquisition of data; YJC, HCC, Acquisition of data, Analysis and interpretation of data, Drafting or revising the article; LCH, Analysis and interpretation of data, Contributed unpublished essential data or reagents; MB, Acquisition of data, Contributed unpublished essential data or reagents; VB, RD, AS, DB, Contributed unpublished essential data or reagents; KC, Drafting or revising the article; BD, Conception and design, Drafting or revising the article, Contributed unpublished essential data or reagents; PAR, Conception and design, Analysis and interpretation of data, Drafting or revising the article

### Author ORCIDs

Samira Lakhal-Littleton, http://orcid.org/0000-0001-7797-1567
Benjamin Davies, http://orcid.org/0000-0003-3623-600X

Peter A Robbins, http://orcid.org/0000-0002-4975-0609

## Ethics

Animal experimentation: All animal procedures were compliant with and approved under the UK Home Office Animals (Scientific Procedures) Act 1986 (Project Licence number 30/3182). In vivo assessment of cardiac function was carried out under anesthesia (2% Isofluorane) to minimize suffering.

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

**Appendix 1**

## Generation of *Slc40a1* C326Y*fl/fl* mice

A C57BL/6J mouse genomic BAC clone (RP23-183-P22) encompassing the *Slc40a1* gene was used for the generation of the targeting vector. A 12 kb region of the RP23-183-P22 encompassing exons 6–8 and surrounding intronic regions was subcloned by gap-repair into a plasmid containing a diphtheria toxin A chain negative selection cassette. The required C326Y point mutation was introduced into this subclone by site directed mutagenesis. In parallel, a loxP flanked neomycin/kanamycin selection cassette (from PL452 *Liu et al., 2003*) was inserted into the BAC within *Slc40a1* intron six by recombineering and an approximately 5 kb ApaI/PmeI fragment encompassing this insertion together with exons 6 and 7 and flanking introns was subcloned into pBluescript. This plasmid was treated with Cre recombinase (NEB) in vitro to remove the selection cassette, resulting in a plasmid containing a single loxP and an additional SpeI site within *Slc40a1* intron 6. This loxP interrupted sequence was shuttled into the genomic subclone by exchanging an internal NcoI fragment. A portion of *Slc40a1* cDNA corresponding to the coding region encoded on exons 7 and 8, including the 3′ untranslated region was amplified by RT-PCR from cDNA prepared from embryonic mouse heart and inserted into a piece of synthetic DNA (Gene Art) corresponding to intronic and intergenic sequence lying immediately upstream of exon seven and downstream of exon 8, respectively. This minigene assembly was cloned upstream of an FRT-PGK-Neomycin-FRT-loxP selection cassette (from P451 *Liu et al., 2003*) and the two pieces together were inserted into the modified genomic subclone above via a unique SacII site, completing the final targeting vector. The completed targeting vector was linearised with AatII and electroporated into C57BL/6N JM8F6 embryonic stem cells. Following selection in 210 µg/ml G418, recombinant clones were screened by PCR to detect homologous recombination over the 3′ arm. A forward primer (5′-CAATAGCAGGCATGCTGGGGATG-3′) binding within the polyadenylation of the selection cassette was used together with a reverse primer (5′- GGCATCTGCTGTCTGTGAAA- 3′) binding downstream of the 3′ homology arm to amplify a 6.6 kb fragment from correctly recombined clones. Positive clones were examined for correct recombination at the 5′ end by long range PCR using a forward primer (5′- TGTACCATGGATGGGTCCTT-3′) binding upstream of the 5′ homology arm and a reverse primer (5′-TACCGGTGGATGTGGAATGTG- 3′) binding within the PGK promoter driving the neomycin selection. Correctly targeted clones yielded a 9.7 kb amplicon. Incorporation of the 5′ loxP site, was verified by digestion of this 5′ amplicon with SpeI which yielded a 7 kb and 2.7 kb cleavage products in correctly targeted clones. Both 5′ and 3′ screening amplicons were analysed by Sanger sequencing to confirm the junction sequence over the extremities of the homology arms using sequencing primers 5′-AAAGCCCAGGGGTATCTCTC-3′ and 5′-CTCAGCTTGGCTATGTGGTG-3′ for the 5′ and 3′ amplicons respectively. The presence of the 5′ loxP site within the 5′ amplicon was confirmed by Sanger sequencing using sequencing primer 5′- CATGAAGCAGTGGGCATAGA-3′ and the presence of the C326Y mutation was confirmed within the 3′ amplicon using sequencing primer 5′-CACACACACACATATATACATGCAA-3′. Southern blotting using a probe against neomycin was used to confirm that only a single integration event had occurred. Correctly recombined ES cells were injected into albino C57BL/6J blastocysts and the resulting chimeras were mated with albino C57BL/6J females. Successful germline transmission yielded black pups and F1 mice harboring the conditional C326Y knock-in allele were identified using the above screening PCR. F1 heterozygous male mice were bred with C57BL/6J Flp recombinase deleter mice (Tg(ACTB-Flpe)9205Dym (Jax stock 005703)) and offspring were screened for the deletion of the selection cassette using a forward primer (5′-GATATCATCATCGCCCTTTGG- 3′) binding within the 3′ untranslated region of the exon 7–8 minigene and a reverse primer (5′- TTGCATGTATATATGTGTGTGTGTG-3′) binding immediately downstream of the cassette. A 1.4 kb amplicon was obtained from the Flp

deleted conditional C326Y knock-in allele. Heterozygous mice without the selection cassette were then backcrossed with C57BL/6J to remove the Flp transgene prior to onward breeding.

## Aconitase and ETC activities

### Total aconitase I activity assay

Total aconitase I activity was measured as the rate of isomerization of cis-aconitic to isocitrate at 240 nm and 37°C using a spectrophotometer. The assay buffer consisted of 50 mmol/L Tris HCl pH 7.5. The reaction was carried out in a quartz cuvette containing 1 ml assay buffer supplemented with 10 μl 20 nmol/L cis-aconitic acid (Sigma) and 25 μl of protein lysate and read for 2 min at 240 nm against a black containing ddH$_2$O. The activity of aconitase was determined by dividing the gradient of the absorbance over the extinction coefficient (3600) and expressed in nmol/min/mg tissue.

### Complex I activity assay

Mitochondrial complex I activity was measured as the rate of NADH oxidation at 340 nm and 30°C using a spectrophotometer. The assay buffer contained the following: 25 Mmol/L potassium phosphate pH 7.2, 5 mmol/L MgCl$_2$ (Sigma), 0.13 mmol/L NADH (MP Biomedicals, LLP), 65 μM coenzyme Q$_1$ (Sigma), 250 mg fatty acid free BSA (Sigma), 200 μg antimycin A (Santa Cruz). The reaction was carried out in 1 ml assay buffer supplemented with 20 μl of protein lysate and read for 1 min at 340 nm against a blank containing ddH$_2$O. As a negative control, 20 μl of 10 mmol/Lrotenone (Sigma) was added to a new reaction and the inhibited rate measured for 1 min. The activity of complex I was determined by dividing the gradient of the absorbance over the extinction coefficient (6810) and expressed in nmol/min/mg tissue.

### Complex II activity assay

Mitochondrial complex II activity was measured as the rate of succinate-dependent reduction of dichlorophenolindophenol (DCIP) at 600 nm and 30°C using a spectrophotometer. The assay buffer contained the following: 25 mol/L potassium phosphate pH 7.2, 5 mmol/L MgCl$_2$, and 2 mmol/L sodium succinate dibasic hexahydrate (Sigma). The reaction was carried out in 985 μl of assay buffer supplemented by 10 μl of buffer containing 2 μg antimycin, 2 μg rotenone, 50 μmol/L DCIP (Sigma) followed by addition of 65 μmol/L CoQ$_1$ and 20 μl protein lysate and read for 2 min at 600 nm against a blank containing ddH$_2$O. As a negative control, 20 μl of 10 mmol/L sodium malonate (Sigma) was added to a new reaction and the inhibited rate measured for 2 min. The activity of complex II was determined by dividing the gradient of the absorbance over the extinction coefficient (21000) and expressed in nmol/min/mg tissue.

### Complex III activity assay

Mitochondrial complex III activity was measured as the rate of reduction of cytochrome $c^{3+}$ at 550 nm at 30°C using reduced ubiquinol as an electron acceptor using a spectrophotometer. To prepare ubiquinol, 10 mg of decylubiquinone (Sigma) was dissolved in 312.5 μl absolute ethanol to give a final concentration of 100 mmol/L. An aliquot of 100 μl decylubiquinone working solution was further diluted in 900 μl ethanol and acidified to pH 2 with a 6 mol/L HCl solution. The ubiquinone was then reduced with a pinch of sodium borohydrate (Sigma) and 1 ml of ddH$_2$O was added to stop the reaction. Excess sodium borohydrate was allowed to settle and the sample centrifuged briefly to separate the sodium borohydrate precipitate from the reduced ubiquinol. Using a pH indicator paper, the pH of the ubiquinol was verified to be at pH two before using in the activity assay. The reaction was carried in 1 ml of

assay buffer (50 mM potassium phosphate pH 7.2, 3 mmol/L sodium azide (Sigma), 1.5 μmol/L rotenone, and 50 μmol/L cytochrome c from bovine heart [Sigma]), supplemented with 5 μl reduced ubiquinol and 20 μl protein lysate and read for 1 min at 550 nm against a blank containing ddH$_2$O. As a negative control, 10 μl of 20 mmol/L antimycin A was added to a new reaction and the inhibited rate measured for 1 min. As further negative control, a reaction containing protein only or reduced ubiquinol only was also measured for 1 min. The activity of complex III was determined by dividing the gradient of the absorbance over the extinction coefficient (19100) and expressed in nmol/min/mg tissue.

## Complex IV activity assay

Mitochondrial complex IV activity was measured as the rate of oxidation of cytochrome c$^{2+}$ at 550 nm at 30°C using a spectrophotometer. To prepare the reduced cytochrome c$^{2+}$, a dialysis tubing (Viking 7000/1) of appropriate length was hydrated for 30 min in 1 L ddH$_2$O supplemented with 20 g sodium carbonate and 0.372 g EDTA at 80°C. A solution of 100 mg of cytochrome c from bovine heart and 10 mg sodium ascorbate dissolved in 10 ml potassium phosphate (0.1 mol/L, pH 7.0) was added to the tubing and dialysed against 1 L of 0.1 mol/L potassium phosphate buffer for 24 hr at 4°C. The phosphate buffer was exchanged three times, every 8 hr. The redox state of the synthesized reduced cytochrome c was verified by measuring the absorbance spectra between wavelengths of 500 and 600 nm, in the presence or absence of 100 μmol/L potassium ferricyanide (Sigma) and compared against the oxidized cytochrome c. The reaction was carried out in 1 ml assay buffer (10 mmol/L potassium phosphate pH 7.0 and 0.50 μmol/L reduced cytochrome c) supplemented with 5 μl protein lysate and read for 3 min at 550 nm against a blank of assay buffer supplemented by 100 μmol/L potassium ferricyanide. As a negative control, 10 μl of 10 mmol/L sodium azide was added to a new reaction and the inhibited rate measure for 3 min. The first-order rate constant (k) was calculated as previously described **Brooks et al. (2000)**. Briefly, the natural logarithm was taken for the absorbances at four time points t = 0, 60, 120, 180 s and the difference for each pair of time points determined (t(60)-t(0), t(120)-t(60), t(180)-t(120)). The average of these differences were taken to be k and the activity expressed in nmol/min/mg tissue.

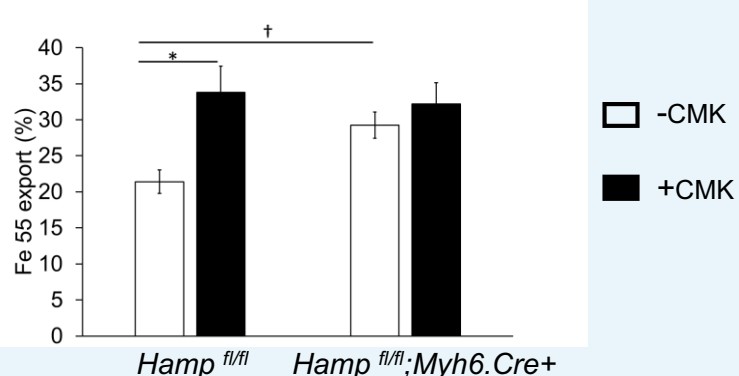

**Appendix 1—figure 1.** Effect of Furin inhibitor on iron export in cardiomyocytes. Fe55 efflux measured in primary adult cardiomyocytes from Hamp *fl/fl* and Hamp *fl/fl;Myh6.Cre+* mice following culture in control medium (-CMK) or medium containing Furin inhibitor (+CMK) for 2 hr. n = 3. values are plotted as mean ± SEM. *p=0.027. †p = 0.024.

Biochemistry | Cell Biology

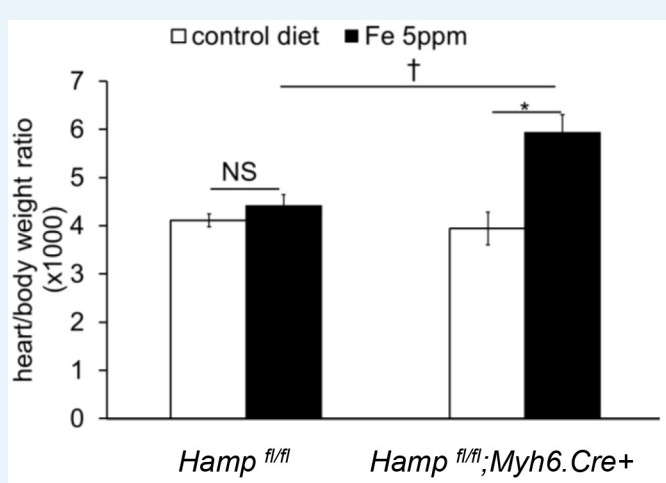

**Appendix 1—figure 2.** Effect of genotype on hypertrophic response to iron-deficient diet. Heart/body weight ratio in Hamp*fl/fl*;*Myh6.Cre+* mice and littermate Hamp*fl/fl* controls provided a control diet (Fe 200 ppm) or an iron-deficient diet (Fe 2–5 ppm) from weaning for six weeks. n = 5 per group, NS=not significant. *p=0.013, †p = 0.04. Values are shown as mean ± SEM.

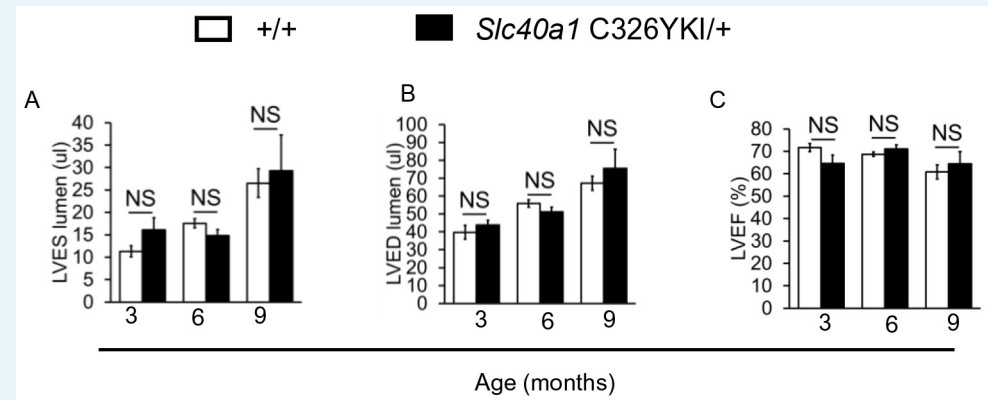

**Appendix 1—figure 3.** Normal cardiac function in ubiquitous *Slc40a1* C326Y KI mice. (**A–C**) Cine MRI measurements of LV lumen, at end-systole (LVES), end-diastole (LVED), and of ejection fraction (LVEF) in *Slc40a1 C326Y*KI/+ mice and +/+ littermate controls at three months (n = 7 per group), six months (n = 7 per group) and nine months (n = 7 per group) of age. Values are plotted as mean ± SEM.

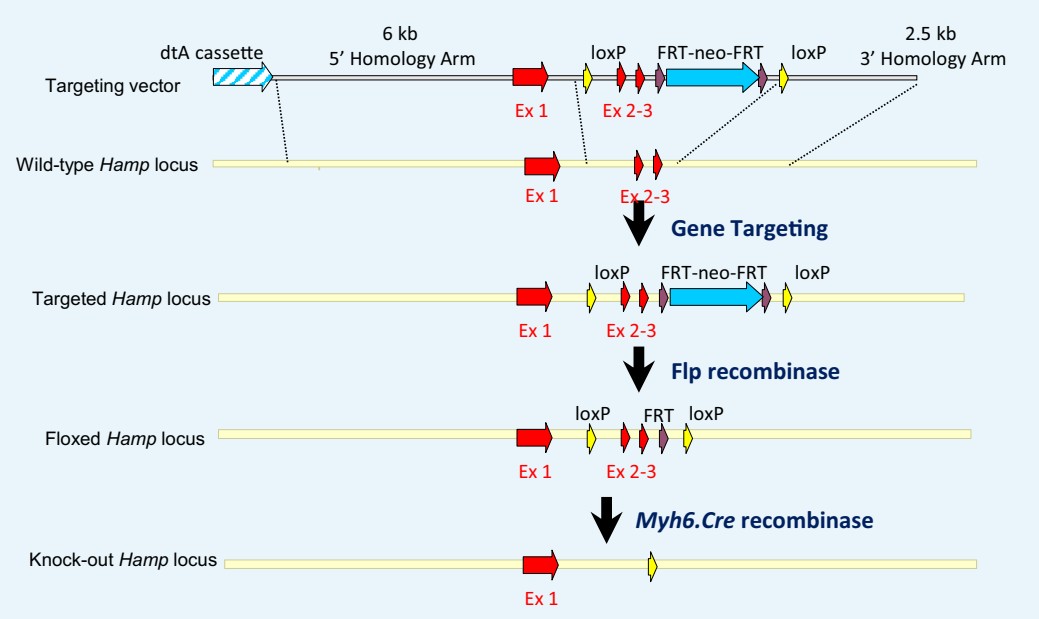

**Appendix 1—figure 4.** Strategy for generation of *Hamp*<sup>fl/fl</sup>;*Myh6.Cre+* mice. A targeting vector was designed to introduce a floxed *Hamp* allele into mouse ES cells, with exons 2 and 3, which encode the majority of the peptide, flanked by LoxP sites. Further breeding with a C57BL/6 Flp recombinase deleter mouse allowed removal of the Neo cassette. Cardiac *Hamp* knockouts were then generated by crossing homozygous *Hamp*<sup>fl/fl</sup> animals with mice transgenic for *Myh6-Cre* recombinase, which is under the control of cardiomyocyte-specific Myosin Alpha Heavy chain six promoter.

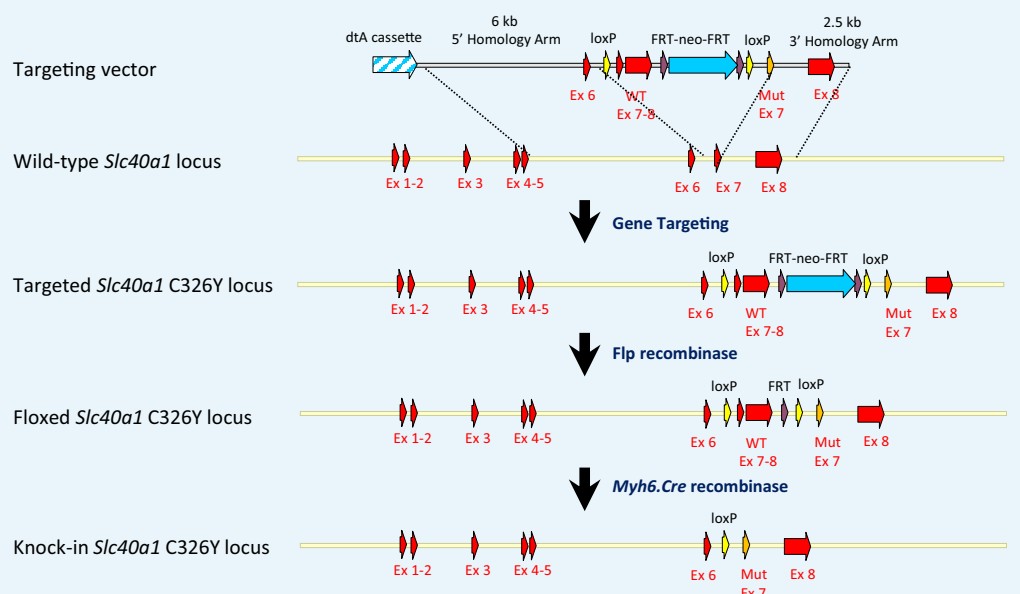

**Appendix 1—figure 5.** Strategy for generation of *Slc40a1 C326Y*<sup>fl/fl</sup>;*Myh6.Cre+* mice. A targeting vector was designed to introduce a floxed *Slc40a1 C326Y* allele into mouse ES cells, containing mutant exon seven and wild type and to delete simultaneously endogenous exons 7 and 8. Further breeding with a C57BL/6 Flp recombinase deleter mouse allowed removal of the Neo cassette. Cardiac *Slc40a1 C326Y* knock-ins were then generated by crossing homozygous *Slc40a1 C326Y*<sup>fl/fl</sup> animals with mice transgenic for Myh6-Cre

recombinase, which is under the control of cardiomyocyte-specific Myosin Alpha Heavy chain six promoter.

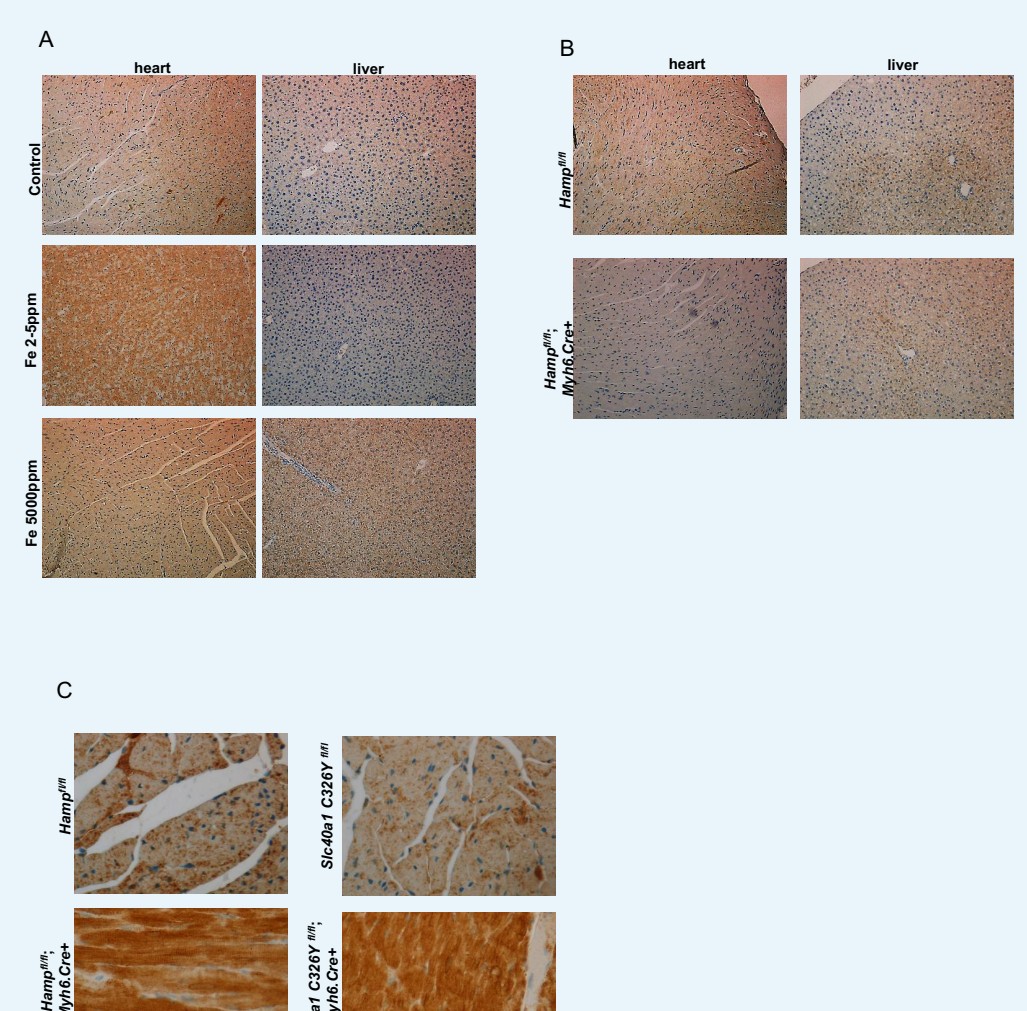

**Appendix 1—figure 6.** Additional images for HAMP and FPN staining. (**A**) Lower magnifications images (X10) for *Figure 1B*. (**B**) Lower magnification images (X10) for *Figure 1F*. (**C**) Higher magnification images (X40) for *Figure 3A and C*.

