## [Decision Letter]

Thank you for submitting your article "An essential cell-autonomous role for hepcidin in cardiac iron homeostasis" for consideration by *eLife*. Your article has been favorably evaluated by Naama Barkai (Senior Editor) and three reviewers, one of whom is a member of our Board of Reviewing Editors. The reviewers have opted to remain anonymous.

The reviewers have discussed the reviews with one another and the Reviewing Editor has drafted this decision to help you prepare a revised submission.

Summary:

The reviewers found the manuscript to have merit and potentially to make an important contribution to understanding iron homeostasis at the tissue level. For the first time, the authors tested the conjecture that tissue hepcidin production may contribute to local tissue-level iron regulation.

However, a number of important issues were raised by the reviewers including the need to be stricter in the methodology used and its description. In particular, the reviewers were seeking further evidence that the data on hepcidin expression and regulation were robust and that bioactive hepcidin was being detected, with hepcidin KOs as controls. Reviewers were reluctant to accept immunohistochemical staining as quantification of bioactive secreted hepcidin. Furthermore, iron efflux experiments need to be described in more detail. Tissue heterogeneity including the possible presence of macrophages may be an issue to resolve. Any further evidence that can be provided on the regulation of hepcidin would strengthen the paper.

I have summarised the major issues below:

1) Subsection “Hamp expression and regulation in the heart”, first paragraph and Figure 1: The semiquantitation of hepcidin by immunohistochemical methods has serious liabilities because hepcidin is a secreted protein and because antibodies can detect functional and nonfunctional hepcidin peptide, including precursor and degraded forms. The conversion of propeptide to active peptide is a step that could be regulated by iron and this may not be detected by the method used. The Methods section indicates that it was a rabbit polyclonal anti-mouse antibody from Abcam, but it does not say which antibody from Abcam was used. Several listed on Abcam's website were prepared against human peptides but have anti-mouse reactivity; one was prepared against mouse antigen and recognizes both Hamp1 and Hamp2 in mice. Are the authors certain that the antibody is recognizing the hepcidin-25 peptide encoded by Hamp1, and not something else? If they are detecting either pro-hepcidin or Hamp2 the conclusions of the paper might be very different. Also, because hepcidin functions as an extracellular peptide, it is difficult to draw conclusions from intracellular antibody staining. Specifically: a) It is not clear which hepcidin forms are stained and where they are located. Does the antibody detect intracellular peptide which is mostly propeptide? Fluorescent confocal microscopy could help with localization. b) No evidence is provided that cardiac myocytes can generate the functional 25 amino acid hepcidin. The processing of peptide hormones to bioactive forms requires prohormone convertases which are expressed in a tissue-specific manner.

2) Subsection “Hamp expression and regulation in the heart”, first paragraph and Figure 1: The detection of hepcidin secreted by cardiomyocytes in culture needs to be validated better because the concentrations are very low, and because the post-mRNA effect of DFO and FAC on hepcidin secretion is unprecedented. This experiment is crucial for understanding the function of local control by hepcidin-ferroportin. It is therefore necessary to be as careful as possible here. Controls should include hepcidin KO cardiomyocytes. At these low peptide concentrations, adsorption of peptide to plastic surfaces causes major losses and these could be affected by DFO and FAC treatment. Neither is it clear which forms of hepcidin are released into the media. Finally, primary hepatocytes within less than 24h in culture lose their ability to physiologically regulate hepcidin in response to iron, and this can result in delayed paradoxical suppression of hepcidin mRNA by FAC. If this happens with cardiomyocytes it could lead to artefacts. The regulation of hepcidin mRNA by FAC and DFO should therefore be checked as a function of time in culture.

3) To confirm the equivalence of Hamp KO and ferroportin C326Y the survival curves need to be compared. This is a hard endpoint that has produced a surprise previously (e.g. the marked difference in survival between global hepcidin KO and global ferroportin C326S). If data exists on the survival of Ferroportin C326Y transgenic mice, it should be presented.

4) It would be relatively straightforward for the authors to examine the regulation of cardiac Hamp using hemojuvelin floxed (Hfe2 fl/fl) mice (Chen W et al., Blood 2011) or other approaches, as regulation of hepatic Hamp expression is well understood. Hfe2 is expressed at high levels in the heart and might be relevant there as it is in the liver. The paper would be much stronger if there was some indication of the mechanism of hepcidin regulation in cardiomyocytes.

5) The authors refer to other animal models of cardiac iron deficiency, but they don't really distinguish between those caused by dietary iron deprivation, which would have the confounding existence of iron deficiency anemia, and a heart-specific Tfrc knockout model in which iron deficiency is restricted to the heart (Xu W et al., Cell Reports 2015, ref 31). The heart-specific Tfrc knockout was shown to have many characteristics that are similar to those described in this paper. One important difference, however, was in the levels of electron transport chain complexes found in untreated and iron treated control and cardiac Hamp-/- mice (Figure 4—figure supplement 2). Xu et al. reported decreases in the bands representing iron-containing ETC complexes using the same OXPHOS blotting approach, as would be expected in iron deficient cells. Those representative proteins are labile when the complexes do not form properly. How do the authors explain their unexpected results, particularly considering the decreased complex activity shown in Figure 4?

6) Several experiments were carried out using isolated adult mouse cardiomyocytes. How did the authors insure that the cell populations were not contaminated with other cell types, particularly macrophages, which might confound the results?

7) The iron efflux experiments shown in Figure 3 appear to be novel because there is no reference provided. Is this correct? For that reason it would be very helpful to know more about them, to assess their validity. Apparently, cells were loaded with non-transferrin bound 55Fe by a 30 minute incubation. How much iron was taken up under these conditions? Why was non-transferrin bound iron used instead of Fe-Tf, particularly considering the results described in Xu et al., 2015 indicating that the transferrin cycle is very important for cardiomyocyte iron uptake? What measures were undertaken to ensure that the PBS washes effectively removed non-assimilated radiolabeled iron? How many times were the experiments repeated, and how variable were the results? Why do the authors think the FpnC326Y mutant had higher iron efflux compared to Hamp-/- cardiomyocytes in the absence of Hamp?

8) Figure 1, panel A – this appears to be mRNA expression but this is not stated. If it is, how was it normalized? In panel B, as the authors pointed out, it is quite surprising that there appears to be most anti-Hamp staining under iron deficient conditions. If this is truly physiologically important staining (see point 2 above) then it becomes even more important to understand how Hamp expression is regulated (see point 3 above). Hamp is typically measured extracellularly, rather than intracellularly because it is a secreted protein; a defect in pre-protein processing might help explain the discrepancy between panels B and C in this figure. Panel D is particularly important – how many times was this experiment replicated? How reliable is the assay kit (the methods indicate that it is from "uscnk" but I don't know what that is – please specify), and is it specific for Hamp1 rather than Hamp2?

9) The heart sections in Figure 2 are suboptimal. It is particularly difficult to discern anatomy in the fl/fl section. In Figure 2, the control cardiomyocytes were much smaller than those reported in Figure 1 of Xu et al., 2015, and the difference in cardiomyocyte surface area between controls and mutants was much larger. Do the authors have any idea what this difference might be due to?

[Editors' note: further revisions were requested prior to acceptance, as described below.]

Thank you for resubmitting your article "An essential cell-autonomous role for hepcidin in cardiac iron homeostasis" for consideration by *eLife*. Your article has been favorably evaluated by Naama Barkai (Senior Editor) and a Reviewing Editor.

The reviewers have discussed the reviews with one another and the Reviewing Editor has drafted this decision to help you prepare a revised submission.

The reviewers generally did not think that the authors use of tissue immunostaining to quantitate a secreted protein is bullet-proof but accept that the authors have enough additional overlapping evidence to give more confidence that the main conclusions are right. They have put together a strong case for hepcidin and ferroportin interacting in an autocrine/paracrine manner in the heart. The genetic argument from the equivalent pathology and similar survival of cardiomyocyte FpnC326Y (cardiomyocyte hepcidin resistance) and cardiomyocyte hepcidin KO is particularly appealing. The autocrine/paracrine function of hepcidin and ferroportin is an old conjecture but this is the first evidence for it.

However, the following is required for acceptance:

1) In the rebuttal there is a statement that the reviewers did not agree with. The cardiac-specific Tfr1 KO mice are not embryonic lethal – they live into the second postnatal week. The authors should be aware of this and modify the text if needed.

2) One reviewer did not believe that the authors had fully explained the outstanding issues with their OXPHOS blot – they thought it was poor quality with much more animal to animal variability than there should be. If possible, the authors should repeat this blot in order to show more consistency between replicates. Alternatively, the authors should delete this western.

---

## [Author Response]

[…]

*1) Subsection “Hamp expression and regulation in the heart”, first paragraph and Figure 1: The semiquantitation of hepcidin by immunohistochemical methods has serious liabilities because hepcidin is a secreted protein and because antibodies can detect functional and nonfunctional hepcidin peptide, including precursor and degraded forms. The conversion of propeptide to active peptide is a step that could be regulated by iron and this may not be detected by the method used. The Methods section indicates that it was a rabbit polyclonal anti-mouse antibody from Abcam, but it does not say which antibody from Abcam was used. Several listed on Abcam's website were prepared against human peptides but have anti-mouse reactivity; one was prepared against mouse antigen and recognizes both Hamp1 and Hamp2 in mice. Are the authors certain that the antibody is recognizing the hepcidin-25 peptide encoded by Hamp1, and not something else? If they are detecting either pro-hepcidin or Hamp2 the conclusions of the paper might be very different. Also, because hepcidin functions as an extracellular peptide, it is difficult to draw conclusions from intracellular antibody staining. Specifically: a) It is not clear which hepcidin forms are stained and where they are located. Does the antibody detect intracellular peptide which is mostly propeptide? Fluorescent confocal microscopy could help with localization. b) No evidence is provided that cardiac myocytes can generate the functional 25 amino acid hepcidin. The processing of peptide hormones to bioactive forms requires prohormone convertases which are expressed in a tissue-specific manner.*

Hepcidin is synthetized as an 84 amino acid prepropeptide, containing an N-terminal ER targeting signal sequence (aa 1-24), a proregion (pro) with a consensus Furin cleavage site (aa 25-59) and a C-terminal bioactive iron-regulatory mature peptide (aa 60-84). The antibody used for immunohistochemical detection of hepcidin in the heart is ab30760 from Abcam, which was raised against the human mature peptide (aa60-84). We now include this information in the Methods section of the revised manuscript.

The reviewers’ concerns over the specificity of this antibody are ones that we also had, and consequently we did carry out appropriate control experiments. We now include the results of those control experiments in the revised manuscript. First, we showed that staining with this antibody is absent in the heart of cardiac-specific hamp-1 knockouts (Figure 1), despite Hamp-2 expression being unaltered in these hearts compared to controls (Figure 1—figure supplement 5). These data demonstrate that this antibody detects HAMP1 and not HAMP2. Second, we showed that staining in the heart of wild-type animals is completely abrogated by the use of the mature peptide (hepcidin-25, Abcam ab 31875) as a blocking peptide (Figure 1—figure supplement 5). In hepatocytes, it has been shown that the propeptide can also be secreted, and that this propeptide is detectable by antibodies raised against the mature peptide (Rouault and Klausner, 1997). Thus, in theory our antibody detects not only the mature secreted peptide but also the propeptide, and even demonstration of extracellular staining by immunohistochemistry would not constitute formal evidence for the presence of the active mature peptide. We do, however, provide direct evidence for the secretion of an active hepcidin by cardiomyocytes. First, in our in vitro experiments, where we detect hepcidin by ELISA in supernatants of primary cardiomyocytes from control animals (original data). We reproduced this finding in a second independent experiment (new Figure 1), and further show that hepcidin is greatly reduced or undetectable in supernatants of cardiomyocytes from cardiac Hamp-1 knockout animals. We now include these data as Figure 1—figure supplement 4. Second, we also provide evidence that the secreted peptide is active, and this is discussed below.

The reviewer is right to point out that the release of the bioactive hepcidin mature peptide requires prohormone convertases, which may or may not be present in the heart. Furin is the prohormone convertase required for release of hepatic bioactive mature hepcidin peptide. In hepatocytes, it has been shown that Furin inhibition leads to decreased release of the active mature peptide and increased release of the larger inactive propetide, and that both of these protein species are detected by assays targeting the mature peptide (Rouault and Klausner, 1997). Furin is also highly expressed in the heart (Beaubien et al., 1995), raising the intriguing possibility that it may be involved in the production of the bioactive hepcidin mature peptide from cardiomyocytes. To test this hypothesis, we performed additional experiments using the Furin inhibitor Decanoyl-Arg-Val-Lys-Arg-chloromethylketone trifluoroacetate salt(CMK). These experiments showed that while Furin inhibition under iron replete conditions did not decrease secreted HAMP levels (consistent with the release of the propeptide as reported in hepatocytes) (Valore and Ganz, 2008), it nonetheless increased iron export from primary cardiomyocytes. Furthermore, the effect of Furin inhibitor on iron export was only seen in Hamp fl/fl and not in Hamp fl/fl Myh6.Cre+ cardiomyocytes. We include these data in Figure 6. Together, these results provide evidence that cardiomyocytes release an active form of hepcidin in a Furin-dependent manner.

Furthermore, we found that increased HAMP release from DFO-treated cardiomyocytes depended on Furin. These data are added to Figure 1. We also found that Furin itself was upregulated by iron deficiency in vitro and in vivo. We include these data as Figure 2. These findings are consistent with the previously reported regulation of Furin by iron deficiency through Hypoxia-inducible factors HIFs (Silvestri, Pagani and Camaschella, 2008). They suggest that differential regulation of Furin by iron may explain the divergent effects of iron on Hamp transcript and HAMP protein.

*2) Subsection “Hamp expression and regulation in the heart”, first paragraph and Figure 1: The detection of hepcidin secreted by cardiomyocytes in culture needs to be validated better because the concentrations are very low, and because the post-mRNA effect of DFO and FAC on hepcidin secretion is unprecedented. This experiment is crucial for understanding the function of local control by hepcidin-ferroportin. It is therefore necessary to be as careful as possible here. Controls should include hepcidin KO cardiomyocytes. At these low peptide concentrations, adsorption of peptide to plastic surfaces causes major losses and these could be affected by DFO and FAC treatment. Neither is it clear which forms of hepcidin are released into the media. Finally, primary hepatocytes within less than 24h in culture lose their ability to physiologically regulate hepcidin in response to iron, and this can result in delayed paradoxical suppression of hepcidin mRNA by FAC. If this happens with cardiomyocytes it could lead to artefacts. The regulation of hepcidin mRNA by FAC and DFO should therefore be checked as a function of time in culture.*

As suggested by the reviewer, we have performed additional experiments with primary cardiomyocytes from cardiac hamp1-ko cells, and found that levels of HAMP proteins were greatly reduced or undetectable in their supernatants. We include this in Figure 1—figure supplement 4. Furthermore, the concentrations detected in the supernatants are well within the detection range (4.94-400pg/mL) of the ELISA kit used (Uscn Life Science Inc. #E91979Mu). The reviewer was also concerned that in vitro treatments (DFO and FAC) may affected detection of the hepcidin peptide by this ELISA assay. To guard against the possibility of differential absorption resulting from the presence of DFO or FAC, we compared the standard curves obtained by dilution of the peptide standard in a) standard diluent provided in the kit, b) unconditioned cardiomyocyte growth medium, c) unconditioned cardiomyocyte growth medium with 100uM DFO, and d) unconditioned cardiomyocyte growth medium with 500uM FAC. We found no significant differences between the 4 standard curves. We include these data as Figure 1—figure supplement 6. This demonstrates that the presence of DFO or FAC does not affect the detection of the hepcidin peptide by this ELISA kit.

The reviewer was concerned that physiological control of hepcidin was lost in cardiomyocytes due to the length of culture. First, both in the original and in the additional experiments performed herein, cardiomyocytes were used within 2 hours of harvest. Additionally, as suggested by the reviewer, we carried out a timecourse of treatment with FAC and DFO, and now present these data in the manuscript (new Figure 1 for Hamp transcript and new Figure 1 for HAMP in supernatants). Overall, the direction of change in response to DFO and FAC, both for hepcidin transcript and protein are maintained over the 24 hour period and are consistent with the single-timepoint data presented in the original manuscript prior to revision.

*3) To confirm the equivalence of Hamp KO and ferroportin C326Y the survival curves need to be compared. This is a hard endpoint that has produced a surprise previously (e.g. the marked difference in survival between global hepcidin KO and global ferroportin C326S). If data exists on the survival of Ferroportin C326Y transgenic mice, it should be presented.*

We now provide those data as new Figure 3. As for Hamp fl/fl;Myh6.Cre+ mice, Slc40a1 fl/fl;Myh6.Cre+ animals also have greater mortality compared to their littermate controls with mean survival of 32 weeks.

*4) It would be relatively straightforward for the authors to examine the regulation of cardiac Hamp using hemojuvelin floxed (Hfe2 fl/fl) mice (Chen W et al., Blood 2011) or other approaches, as regulation of hepatic Hamp expression is well understood. Hfe2 is expressed at high levels in the heart and might be relevant there as it is in the liver. The paper would be much stronger if there was some indication of the mechanism of hepcidin regulation in cardiomyocytes.*

HJV is one of many putative regulators of cardiac hepcidin. As it happens, muscle-specific HJV KO mice have increased rather than decreased cardiac hepcidin expression (Kostas Pantopoulos, personal communication). Thus, it appears that both with respect to HJV and to iron, cardiac hepcidin is regulated differently from liver hepcidin. Substantive additional studies are required to explore the role of known regulators of hepatic hepcidin in the regulation of cardiac hepcidin. We consider such studies beyond the scope of this revision as the work would constitute an independent manuscript.

*5) The authors refer to other animal models of cardiac iron deficiency, but they don't really distinguish between those caused by dietary iron deprivation, which would have the confounding existence of iron deficiency anemia, and a heart-specific Tfrc knockout model in which iron deficiency is restricted to the heart (Xu W et al., Cell Reports 2015, ref 31). The heart-specific Tfrc knockout was shown to have many characteristics that are similar to those described in this paper. One important difference, however, was in the levels of electron transport chain complexes found in untreated and iron treated control and cardiac Hamp-/- mice (Figure 4—figure supplement 2). Xu et al. reported decreases in the bands representing iron-containing ETC complexes using the same OXPHOS blotting approach, as would be expected in iron deficient cells. Those representative proteins are labile when the complexes do not form properly. How do the authors explain their unexpected results, particularly considering the decreased complex activity shown in Figure 4?*

With regards to the description of animal models of cardiac iron deficiency, we have now amended the text to distinguish between the various models, as suggested by the reviewer. With regards to the levels of ETC complexes detected by OXPHOS, we too were initially surprised that we did not detect a reduction as was seen in cardiac TfR knockout mice (Xu et al., 2015).However, this becomes less surprising when you consider the relative severity of cardiac iron deficiency, which is reflected in the fact that cardiac-specific TfR KO are embryonic lethal whereas cardiac-specific hepcidin knockouts have a mean survival of 28 weeks. In this context, it is important to consider the mechanisms through which iron deficiency impinges on the ETC. These mechanisms include not only limiting iron availability for complex assembly and for enzymatic activity of assembled complexes, but also reducing the levels of mitochondrially-encoded proteins. Indeed, the degree of iron deficiency (and iron excess) have been shown to correlate with damage to mitochondrial DNA (Walter et al., 2002), and many of the proteins in ETC complexes are encoded by mitochondrial genes. Thus, depending on the degree of iron deficiency in the cardiomyocytes, different mechanisms may come into play, meaning that complex assembly or activity could still be disrupted when levels of individual proteins in the complex are still maintained.

*6) Several experiments were carried out using isolated adult mouse cardiomyocytes. How did the authors insure that the cell populations were not contaminated with other cell types, particularly macrophages, which might confound the results?*

The reviewer is right to point out that macrophages are known to be present in the cardiac tissue. We cannot exclude the presence of macrophages in our cardiomyocyte culture. However, specifically with respect to hepcidin production in the supernatants, we can be confident that cardiomyocytes are the source of the hepcidin, because very little hepcidin was detected in supernatants of cardiac-specific hepcidin knockouts (where cardiomyocytes but not macrophages have deletion of the hepcidin gene). We include these data in Figure 1—figure supplement 4.

*7) The iron efflux experiments shown in Figure 3 appear to be novel because there is no reference provided. Is this correct? For that reason it would be very helpful to know more about them, to assess their validity. Apparently, cells were loaded with non-transferrin bound 55Fe by a 30 minute incubation. How much iron was taken up under these conditions? Why was non-transferrin bound iron used instead of Fe-Tf, particularly considering the results described in Xu et al., 2015 indicating that the transferrin cycle is very important for cardiomyocyte iron uptake? What measures were undertaken to ensure that the PBS washes effectively removed non-assimilated radiolabeled iron? How many times were the experiments repeated, and how variable were the results? Why do the authors think the FpnC326Y mutant had higher iron efflux compared to Hamp-/- cardiomyocytes in the absence of Hamp?*

The method for measuring iron efflux used in our experiments is not novel, but adapted from the method used by Mckie et al. 2000, which was the first publication to describe the function of FPN in iron efflux. We now include this reference in the text. This method does not enable measurement of absolute amounts of iron flux, but rather measurement of iron efflux as a percentage of the total iron taken up during incubation with the uptake solution; percentage efflux= 100 x Fe55 in efflux solution /(Fe55 in efflux+ Fe55 remaining in cells). The readouts are obtained using a scinitillation counter and are expressed as counts per minute. Count rate does not universally equate to dose rate, and there is no simple universal conversion factor. Any conversions are instrument-specific.

The reviewer also questions the use of non-transferrin bound Fe55 (as in Mckie et al., 2000) instead of transferrin-bound Fe55 for loading cells with iron. The rationale for this is to circumvent any confounding effects of differential iron uptake between cells of different genotypes due to differing levels of TfR1 expression. Indeed, we know that TfR1 expression is higher in cardiac-hepcidin knockout cells and in Slc40a1 C326Y knock-in cells than in their respective controls (Figure 3). While TfR1 is the dominant route of iron uptake into cardiomyocytes, it is known that NTBI also enters cardiomyocytes via L and T-type calcium channels.

To minimise carryover of Fe55, cells were washed with PBS 3 times before addition of efflux medium. Efflux measurements were carried out in three independent experiments, with each experiment containing three biological replicates (3 wells of cardiomyocytes) for each genotype and condition. The range between values obtained from three biological replicates constituted between 0.07and 0.21 of the value of mean. Thus, the results were sufficiently accurate to make meaningful comparisons between genotypes and treatment conditions within each experiment, and the outcome of those comparisons was reproducible between experiments.

The reviewer raises an interesting question relating to the levels of iron export being higher in Slc40a1 C326Y knock-in cardiomyocytes than in Hamp knockout cardiomyocytes (p=0.014). We too were initially surprised at this result. One possible explanation is that FPN levels in Slc40a1 C326Y knock-in cardiomyocytes are higher than in Hamp KOs cardiomyocytes, due to the fact that FPN in the former setting is completely insensitive to any hepcidin, while FPN in the latter may still be subject to some regulation by hepcidin from non-cardiomyocyte sources.

*8) Figure 1, panel A – this appears to be mRNA expression but this is not stated. If it is, how was it normalized? In panel B, as the authors pointed out, it is quite surprising that there appears to be most anti-Hamp staining under iron deficient conditions. If this is truly physiologically important staining (see point 2 above) then it becomes even more important to understand how Hamp expression is regulated (see point 3 above). Hamp is typically measured extracellularly, rather than intracellularly because it is a secreted protein; a defect in pre-protein processing might help explain the discrepancy between panels B and C in this figure. Panel D is particularly important – how many times was this experiment replicated? How reliable is the assay kit (the methods indicate that it is from "uscnk" but I don't know what that is – please specify), and is it specific for Hamp1 rather than Hamp2?*

In Figure 1, panel A depicts mRNA expression of hepcidin. We have amended the corresponding figure legend to make this clear. Expression was measured using Taqman multiplex gene expression assays, where hepcidin and the house keeping gene (b-actin) were measured in the same sample. A δ CT (CT for hepcidin-CT for b-actin) value was generated for each sample. Subsequently δ CT values for each sample were normalised to the average δ CT of the control samples (hearts under control diet), to obtain a δ CT value. Relative expression was then calculated as 2^-δ δ CT^. The reviewer’s questions relating to the discrepancy between hepcidin transcript and protein, the possible role of protein processing and the measurement of hepcidin in supernatants are addressed in responses to points 1 and 2 above.

*9) The heart sections in Figure 2 are suboptimal. It is particularly difficult to discern anatomy in the fl/fl section. In Figure 2, the control cardiomyocytes were much smaller than those reported in Figure 1 of Xu et al., 2015, and the difference in cardiomyocyte surface area between controls and mutants was much larger. Do the authors have any idea what this difference might be due to?*

We provide better heart sections in Figure 2bB In terms of cardiomyocyte size, differences in reported cardiomyocyte size between our data (range between 25 and 37 um2) and those of reported by Xu et al^4^ (range between 45 and 60 um2) in control animals may be due to any number of reasons including: a) the manual nature of quantitation of cardiomyocyte size, which requires drawing around individual WGA-stained cardiomyocytes, which is likely to vary between individual users, b) the method for tissue fixing and processing of the cardiac tissue, c)the area of the heart selected for histology and d) orientation of section plane relative to cardiomyocyte axis.

[Editors' note: further revisions were requested prior to acceptance, as described below.]

*The reviewers generally did not think that the authors use of tissue immunostaining to quantitate a secreted protein is bullet-proof but accept that the authors have enough additional overlapping evidence to give more confidence that the main conclusions are right. They have put together a strong case for hepcidin and ferroportin interacting in an autocrine/paracrine manner in the heart. The genetic argument from the equivalent pathology and similar survival of cardiomyocyte FpnC326Y (cardiomyocyte hepcidin resistance) and cardiomyocyte hepcidin KO is particularly appealing. The autocrine/paracrine function of hepcidin and ferroportin is an old conjecture but this is the first evidence for it.*

*However, the following is required for acceptance:*

*1) In the rebuttal there is a statement that the reviewers did not agree with. The cardiac-specific Tfr1 KO mice are not embryonic lethal – they live into the second postnatal week. The authors should be aware of this and modify the text if needed.*

We thank the reviewers for drawing our attention to the erroneous statement that “cardiacTfR1 knockout are embryonic lethal”. In the original response letter, the phenotype of these mice was discussed in relation to our results showing the levels of ETC complexes as detected by the OXPHOS assay (Figure 4—figure supplement 1). As these data have now been removed from the manuscript, so has the erroneous statement in the revised response letter.

*2) One reviewer did not believe that the authors had fully explained the outstanding issues with their OXPHOS blot – they thought it was poor quality with much more animal to animal variability than there should be. If possible, the authors should repeat this blot in order to show more consistency between replicates. Alternatively, the authors should delete this western.*

In relation to the OXPHOS blot showing the levels of ETC complexes (Figure 4—figure supplement 1), we have now run the blot a second time. However, the variability between some of the samples persists. As such, we have opted to remove the blot from the manuscript, and have also removed all text relating to it in the Results, Discussion, Methods and figure legends section. Accordingly, we have also removed any text in our original response letter relating to this particular dataset.